# WASSERSTEIN MOTIFS: NON-DETERMINISTIC ALIGNMENT OF ECOLOGICAL NETWORKS

## ABSTRACT

We study the problem of ecological network (food web) alignment, where we seek to identify structural equivalences among species and uncover backbones of interactions that represent shared functional substructures. These fundamental properties reveal the functional relationships that sustain ecosystems, enabling more accurate predictions of biodiversity responses to environmental change. Existing methods are computationally expensive, not scalable, and hard to interpret ecologically. We provide a first rigorous formalization of food web alignment based on network motifs, and show existing methods popularized in the ecological community are equivalent to minimizing a Fused Gromov-Wasserstein-like cost functional, termed *Wasserstein Motifs*. Moreover, we propose an interpretable and provably correct algorithm that efficiently computes non-deterministic alignments between food webs by leveraging their representation as feature measure networks. As a byproduct, we introduce a novel approach for identifying the non-deterministic backbones of interactions. Experiments on a continental-scale dataset of 129 Sub-Saharan African mammal food webs demonstrate significant gains in accuracy, scalability, and interpretability over state-of-the-art methods. Our results establish a principled bridge between ecological network science and optimal transport, opening avenues for the analysis of complex structured data.

## 1 INTRODUCTION

Ecological networks represent the web of interactions among species within an ecosystem, with nodes denoting individual species and edges indicating interactions, such as predation. A species' ecological role is thus encoded in its network position, and species with similar functions tend to occupy structurally comparable niches within the network (Mora et al., 2018a). Identifying such similarities across ecosystems requires establishing correspondences between species in different networks—a task naturally formulated as a network alignment problem.

Network alignment (NA) aims to identify correspondences between nodes in different networks according to a task-specific cost function. The problem originates from the quadratic assignment problem (QAP) (Koopmans & Beckmann, 1957), which seeks a bijection between node sets that minimizes a non-convex, structure-dependent cost. Classical methods for approximating QAP solutions, such as branch-and-bound and linear relaxations, face severe scalability limitations on large graphs (Burkard, 1984). To overcome these computational barriers, modern approaches map nodes into structure-preserving embeddings, usually reducing NA to tractable linear assignment problems. The central challenge then lies in constructing effective embeddings, which are typically obtained from spectral properties of graph representations (e.g., adjacency or Laplacian matrices) (Goyal & Ferrara, 2018), or learned through models such as graph neural networks (GNNs) (Xu et al., 2019; He et al., 2024).

A pioneering work in ecological network analysis (Mora et al., 2018a) provided an NA formulation based on *network motif distributions* (small, recurring subgraphs) to quantify species similarity and alignment quality. Originally created for social network analysis (Holland & Leinhardt, 1976) and popularized in systems biology and network science (Milo et al., 2002), network motifs have since been adopted widely across the machine learning and data mining communities as local building blocks for complex networks (Benson et al., 2016; Ribeiro et al., 2017; Sankar et al., 2020; Chen et al., 2021; Yu & Gao, 2022). In particular, motif counts and motif-based embeddings have been

utilized as features for graph classification, role discovery, and representation learning, offering a compact and interpretable approach to capture higher-order connectivity beyond pairwise edges. In ecological network analysis, motifs are widely used to capture *local structural information* of food webs, mutualistic systems, and multi-trophic interaction networks (Baiser et al., 2016; Tavella et al., 2022; Cenci et al., 2018; Paulau et al., 2015). Motif structure has been linked to interpretable functional roles of species (e.g., basal producers, intermediate consumers, top predators) and to key ecosystem-level properties such as stability, robustness, and energy flow. Aggregating motif role occurrences for species yields their motif role profiles (Stouffer et al., 2012), which quantify the functional similarity of species across networks. However, the approach in (Mora et al., 2018a) looks for deterministic alignments, i.e., one-to-one correspondences. Moreover, simulated annealing is used to compute alignments, which is slow for large networks and can lead to inconsistent alignments, a problem that is exacerbated by the existence of functionally equivalent species. An alternative motif-based approach is proposed in Almulhim et al. (2019), but it also suffers from the rigidity of deterministic alignments. Deterministic alignments restrict correspondences to one-to-one pairings between species across networks. While such mappings are intuitive and interpretable, they impose rigid constraints that limit ecological applicability. In practice, species frequently exhibit overlapping or redundant functional roles (Stouffer et al., 2012). Enforcing a strict one-to-one correspondence discards these valid alternatives and obscures functional redundancy. These drawbacks restrict the feasibility of deterministic approaches for large or heterogeneous ecological datasets.

Optimal transport (OT) (Cuturi, 2013; Sturm, 2006; Mémoli, 2011) has enabled other NA paradigms. OT-based methods offer superior speed compared to classical methods and enable additional modeling capabilities, as alignments are allowed to be non-deterministic (many-to-many). Moreover, the recently proposed Partial Gromov-Wasserstein (PGW) distance (Bai et al., 2024) has enabled partial non-deterministic alignments, where only a subset of nodes needs to be aligned. In ecological NA domains, partial non-deterministic alignments are crucial for identifying functionally equivalent species, that is, species that play similar roles within a network. In Hung et al. (2025), NA is formulated using the Gromov-Wasserstein (GW) distance, allowing for non-deterministic alignments, but the mass conservation constraint imposed by GW lacks the flexibility to allow species playing unique functional roles to self-align, potentially forcing undesired alignments. This mass conservation constraint is relaxed in the Partial Gromov-Wasserstein distance (Bai et al., 2024) and has been applied for NA (Liu et al., 2020), but the Gromov-Wasserstein cost depends only on nodes' local neighborhood, which is insufficient for characterizing functional species equivalence. This can be mitigated by endowing nodes with a motif-based node embedding and then using a partial version of the Fused Gromov-Wasserstein distance (FGW) (Vayer et al., 2020; Chapel et al., 2020), but the quadratic cost in the FGW still only depends on local connectivity. Ecological network data are scarce because collection is time- and resource-intensive, and inconsistencies across research groups further limit the usable datasets (de Aguiar et al., 2019). Therefore, recent deep learning approaches to network alignment (Xu et al., 2019; Ratnayaka et al., 2024) face limited applicability, since the lack of large-scale homogeneous data hinders their effectiveness on the ecological network alignment task. See Appendix A.2 for a table that summarizes recent network alignment methods.

In this paper, we leverage a motif-based ecological network representation and propose a provably correct algorithm for computing *non-deterministic motif-based* alignments between networks. A schematic overview of the complete Wasserstein Motif Pipeline is provided in Appendix A.9.

*Our main contributions are as follows:*

1. *Mathematical formalization of ecological network alignment:* We provide a first mathematically rigorous formulation for ecological network alignment based on network motifs.
2. *Non-deterministic alignment:* We propose a provably correct algorithm for the computation of non-deterministic food web alignments and the identification of structurally equivalent species.
3. *Backbone identification:* We introduce the notion of the backbone of interactions for non-deterministic alignments.
4. *Numerical Analysis:* We verify that our proposed formulation outperforms baselines in the ecological network alignment task and produces valid backbones on a continental-scale dataset.

Figure 1 illustrates alignments between two toy food webs (red: species A−D, blue: species a−e), with arrows indicating predator→prey interactions. Prior approaches (Mora et al., 2018a; Hung et al., 2025; Bai et al., 2024) produce several inconsistencies: for example, species D is often misaligned with nodes of different trophic roles, and species e is either weakly or spuriously aligned. In contrast, our method consistently aligns A−C with a−c, preserving the top-to-bottom trophic or-

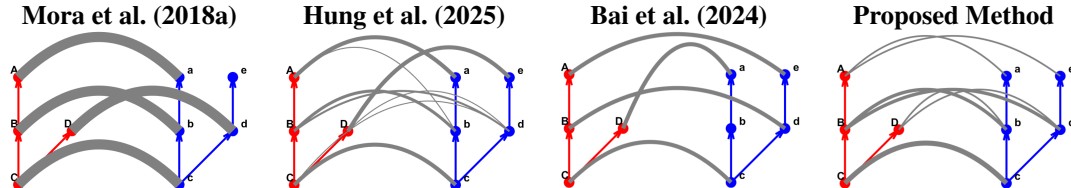

| Mora et al. (2018a) | Hung et al. (2025) | Bai et al. (2024) | Proposed Method |

Figure 1: Alignment between two toy networks. Thicker lines indicate stronger alignment.

dering, while correctly distinguishing between the structurally distinct species D and e. This results in stronger, ecologically coherent correspondences that better capture the functional roles of species across the two networks. See Appendix A.1 for more examples.

**Notations:** Denote $\mathbb{R}$ and $\mathbb{R}_+$ as the set of real and non-negative real numbers, respectively, and $[n] := \{1, 2, \cdots, n\}$ be the set of positive integers up to and including $n$. The cardinality of a set $S$ is denoted as $|S|$, and the vectors of ones and zeros in $\mathbb{R}^d$ by $1_d$ and $0_d$, respectively. Denote the $d$-simplex by $\Delta^{d-1}$ where $\Delta^{d-1} = \{x \in \mathbb{R}_+^d \mid x^\top 1_d = 1\}$, and the uniform discrete measure over the discrete set $S$ as $\nu_S$, i.e., $\nu_S := \sum_{s \in S} \delta_s$, where $\delta_s(x) = 1$ if $x = s$ and 0 otherwise. Let $\hat{\mu}$ denote a finite measure supported by a finite and discrete set $S$. We denote the probability measure induced by $\hat{\mu}$ to be $\mu$, i.e., $\mu : S \to \Delta^{|S|-1}$ is defined as $\mu := \frac{1}{\hat{\mu}(S)} \sum_{s \in S} \hat{\mu}(\{s\})\delta_x$. The element at the $i$-th column and $j$-th row of a matrix $A \in \mathbb{R}^{m \times n}$ is denoted as $A_{ij}$ for all $i \in [m]$ and $j \in [n]$. Assuming an order for the elements in a set $S = \{s_1, \cdots, s_{|S|}\}$, we can identify the measure $\mu_S$ on $S$ with a vector in $\mu \in \mathbb{R}^{|S|}$, where $\mu_i = \mu(\{s_i\})$. The Hadamard product of two matrices $A, B \in \mathbb{R}^{m \times n}$ is denoted as $A \odot B$. The indicator function of a set is defined as $\mathbb{I}_{\mathcal{C}}(x) = 0$ if $x \in \mathcal{C}$, and $\mathbb{I}_{\mathcal{C}}(x) = +\infty$, otherwise. The diagonal matrix formed by a vector $x \in \mathbb{R}^n$ is denoted as $\text{diag}(x) \in \mathbb{R}^{n \times n}$. The number of vertices of a (measure or feature measure) network $\mathcal{X}$ is denoted as $m := |X|$. A finite, unweighted, undirected, (feature) measure network is defined as a tuple $\mathcal{X} = (X, A, \hat{\mu}, \phi)$ consisting of a set of nodes $X = \{x^1, \cdots, x^m\}$, an adjacency matrix $A \in \mathbb{R}^{m \times m}$, a fully-supported measure $\hat{\mu} : X \to \mathbb{R}_+^m$, and a feature vector $\phi : X \to \mathbb{R}^M$.

## 2 NON-DETERMINISTIC FEATURED MEASURE NETWORK ALIGNMENT

Given two featured undirected measure networks $\mathcal{X}_1 = (X_1, A_1, \hat{\mu}_1, \phi_1)$ and $\mathcal{X}_2 = (X_2, A_2, \hat{\mu}_2, \phi_2)$, we seek an alignment matrix $T \in \mathbb{R}_+^{m \times n}$ that balances structural consistency with feature similarity. Inspired by the Fused Gromov–Wasserstein (FGW) loss function (Vayer et al., 2020), we consider the following optimization problem.

$$\min_{T \in \mathcal{T}(\mathcal{X}_1, \mathcal{X}_2)} g(T; C, \alpha, \epsilon) \triangleq \underbrace{(1-\alpha)\langle T, C \rangle}_{\text{zeroth-order similarity}} + \underbrace{\alpha\langle T, A_1(T \odot C)A_2 \rangle}_{\text{first-order similarity}} + \underbrace{\Xi_{\alpha,\epsilon}(T)}_{\text{self-alignment penalty}}, \quad (1)$$

$$\Xi_{\alpha,\epsilon}(T) \triangleq -\epsilon(\alpha\langle T, A_1 T A_2 \rangle + (1-\alpha)\|T\|_1),$$

$$\mathcal{T}(\mathcal{X}_1, \mathcal{X}_2) \triangleq \{T \in \mathbb{R}_+^{m \times n} \mid T 1_n \preceq \hat{\mu}_1 \text{ and } T^\top 1_m \preceq \hat{\mu}_2\},$$

where $\alpha \in [0, 1]$ and $\Xi_{\alpha,\epsilon}$ denote a self-alignment penalty. Moreover, for $x^i \in \mathcal{X}_1$ and $x^j \in \mathcal{X}_2$, we define their feature dissimilarity by $d\left(\phi_1(x^i), \phi_2(x^j)\right)$, where $d : \mathbb{R}^M \times \mathbb{R}^M \to \mathbb{R}_+$, where $d$ is any nonnegative discrepancy. Collecting all pairwise dissimilarities yields the cost matrix $C \in \mathbb{R}_+^{m \times n}$ with entries $C_{ij} := d(\phi_1(x^i), \phi_2(x^j))$, where $i \in [m], j \in [n]$.

Note that a variant of our method using *directed* adjacency matrices has been considered. However, this variant performs less favorably as compared to our formulation. More details and experiments regarding this variant can be found in Appendix A.10.

In (1), the zeroth-order similarity term promotes direct feature matching; note that it is minimized when the alignment assigns values proportional to the cost matrix $C$. The first-order similarity term encourages adjacency relations to be preserved under the alignment; it is minimized when the alignment matrix assigns values to entries where neighbors of nodes are well aligned. They form a "fused" objective that considers feature representations of vertices and network topology between vertices. Moreover, in the absence of regularization, the trivial alignment $T = \mathbf{0}$ minimizes (1), corresponding to no alignment at all. The self-alignment term penalizes trivial self-alignments where

a node is not aligned with any other node. Specifically, for an $\epsilon > 0$, we can define $C_\epsilon := C - \epsilon 1_m 1_n^\top$, and the objective in (1) becomes:

$$g(T; C, \alpha, \epsilon) = \alpha \langle T, A_1(T \odot C_\epsilon)A_2 \rangle + (1 - \alpha)\langle T, C_\epsilon \rangle,$$
$$= \alpha \langle T, A_1(T \odot C)A_2 \rangle + (1 - \alpha)\langle T, C \rangle - \epsilon(\alpha \langle T, A_1 T A_2 \rangle + (1 - \alpha)\|T\|_1).$$

Additionally, note that $\Xi_{\alpha,\epsilon}(T) \propto -\epsilon(\alpha \langle T, A_1 T A_2 \rangle + (1 - \alpha)(\|t_1\|_1 + \|t_2\|_1))$, where the vectors $t_1 := \hat{\mu}_1 - T1_n$ and $t_2 := \hat{\mu}_2 - T^\top 1_m$, are called the self-alignment vectors.

Note that $C$ denotes the base feature dissimilarity matrix (non-negative by construction), while $C_\epsilon$ is a shifted version introduced solely within the optimization problem to enforce the self-alignment regularization term. Hence, although $C$ may contain only non-negative entries, $C_\epsilon$ can include negative values without affecting the interpretation of $C$ itself as a cost matrix.

If we let $\hat{\mu}_1$ and $\hat{\mu}_2$ be probability measures, we call the solution of (1) an optimal *non-deterministic* alignment, or an optimal *deterministic* alignment if $T \in \{0, 1\}^{m \times n}$. Informally, we distinguish whether each unit of mass (or measure) in one network is mapped exclusively to a single counterpart or may be split across multiple counterparts. Deterministic alignments (with probability measures) are thus a special case of non-deterministic alignments where $T$ is restricted to be binary. This distinction mirrors the classical relationship between the Monge and Kantorovich formulations of optimal transport (Villani et al., 2008; Kantorovich, 1942); we relax deterministic pairings into many-to-many alignments (Peyré et al., 2019). Feasible deterministic alignments correspond to a matching in a complete bipartite graph when $\alpha = 0$, which can be solved in polynomial time (Kuhn, 1955). However, once structural terms are incorporated ($\alpha > 0$), the objective no longer admits such a reduction, and the problem becomes a general quadratic assignment problem.

**Proposition 1.** *Let $\mathcal{X}_1 = (X_1, A_1, \nu_{X_1})$, $\mathcal{X}_2 = (X_2, A_2, \nu_{X_2})$ be measure networks with uniform discrete measures over their respective vertex sets. Let $K_{m,n} = (X_1 \cup X_2, E)$ be the complete bipartite graph over their vertex sets. Then, the set of matchings in $K_{m,n}$ is in bijection with the set of deterministic alignments between $\mathcal{X}_1$ and $\mathcal{X}_2$.*

To avoid confusion with the Fused Gromov–Wasserstein (FGW) formulation, we emphasize that our objective is structurally distinct from FGW in both modeling assumptions and objective structure. Classical FGW assumes two metric spaces $(X_1, d_1)$ and $(X_2, d_2)$ and its quadratic term compares all distance pairs through

$$\sum_{i,j,i',j'} L\big(d_1(x_i, x_{i'}), d_2(y_j, y_{j'})\big) T_{ij}T_{i'j'}.$$

In our formulation, structural information is encoded by adjacency. Expanding our first-order term,

$$\langle T, A_1(T \odot C)A_2 \rangle = \sum_{i,j,i',j'} (A_1)_{ii'}(A_2)_{jj'} C_{i'j'} T_{ij}T_{i'j'},$$

reveals a fundamentally different interaction: only actual edges $(A_1)_{ii'} = 1$ and $(A_2)_{jj'} = 1$ contribute, making the structural cost strictly local and feature-aware. Feature dissimilarity $C$ is embedded directly inside this localized structural interaction, coupling motif-role similarity with adjacency in a way that cannot be reproduced by any choice of metrics $d_1, d_2$ or loss $L$ in FGW. For this reason, FGW with motif-based features is not a feature-matched surrogate for our formulation. For more details and numerical experiments comparing motif-based FGW and our formulation, see Appendix A.6.

## 2.1 COMPUTATION OF NON-DETERMINISTIC ALIGNMENTS

From an optimization perspective, Problem (1) in its deterministic variations is a non-convex constrained optimization problem over transport matrices with inequality marginals. Coupled with the integrality constraints, the deterministic alignment problem is a (combinatorial) quadratic assignment problem (Koopmans & Beckmann (1957)). Efficiently solving this problem is out of the scope of this work; we refer the curious reader to Gurobi Optimization, LLC (2024) for a baseline and Shi et al. (2025) for a recent advancement in this direction. Thus, we focus on the *non-deterministic alignment problem*. We propose a variation of the KL Bregman Alternating Projected Gradient (KL-BAPG) algorithm with Dykstra subroutines. This approach preserves non-negativity by design,

achieves a practical balance between accuracy and efficiency, and is supported by strong theoretical guarantees from prior work Li et al. (2023); Benamou et al. (2015).

We propose the following algorithm for the computation of an optimal non-deterministic alignment between $\mathcal{X}_1$ and $\mathcal{X}_2$ (see detailed derivation in Appendix A.3):

$$\hat{T}^{(k)}=P_{\mathcal{C}_1}\big(T^{(k-1)}\odot\exp(-\gamma Q^{(k-1)})\big),\ \text{and}\ T^{(k+1)}=P_{\mathcal{C}_2}\big(\hat{T}^{(k)}\odot\exp(-\gamma Q^{'(k)})\big), \tag{2}$$

$$Q^{(k)}:=\alpha A_1(C_\epsilon\odot T^{(k-1)})A_2+\frac{1}{2}(1-\alpha)C_\epsilon,\ \text{and}\ Q^{'(k)}:=\alpha C_\epsilon\odot(A_1\hat{T}^{(k)}A_2)+\frac{1}{2}(1-\alpha)C_\epsilon,$$

where $\mathcal{C}_1(\mu):=\{T\in[0,1]^{m\times n}\mid T1_n\preceq\mu\}$, $\mathcal{C}_2(\nu):=\{T\in[0,1]^{m\times n}\mid T^\top 1_m\preceq\nu\}$, and $\gamma>0$.

**Assumption 1** (Bounded accumulative asymmetrical error (AAE)). *The iterates generated by Eq. (2) have the following property:* $\sum_{k=0}^\infty(D_h(\hat{T}^{(k)},T^{(k+1)})-D_h(T^{(k+1)},\hat{T}^{(k)}))<\infty$, *where $D_h$ is the Bregman divergence generated by the entropy function $h(X)=\sum_{ij}x_{ij}\log x_{ij}$.*

Assumption 1 is the key condition that allows us to control the non-symmetric part of the Bregman divergence $D_h$ when we build a global Lyapunov function for the alternating scheme. When $h$ is *quadratic*, $D_h$ is symmetric and the asymmetric term in the summand vanishes identically, so Assumption 1 holds trivially (this is precisely the setting treated in the quadratic case (Li et al., 2023, Proposition 3.5)). In the entropic case, the Bregman divergence is non-symmetric, and the differences do not cancel in general; proving that its accumulation is finite couples the geometry of the entropy and the two-block structure of the algorithm. Verifying it analytically for entropic Bregman genera-

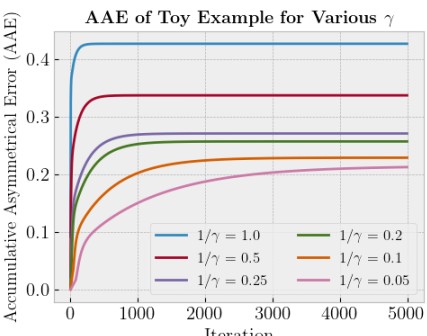

Figure 2: Accumulative Asymmetric Error.

tors is currently an open question. Following the approach in (Li et al., 2023, Figure 3), we empirically verify the result and observe that the cumulative error stabilizes for all step sizes we consider. However, Assumption 1 is a concrete instance of the classical "summable error" conditions that appear in the analysis of inexact proximal and Bregman methods: the descent inequality holds up to an error term, and one assumes that these errors form an absolutely summable sequence in order to apply the Kurdyka–Łojasiewicz framework (Güler, 1992; Attouch et al., 2013; Yang & Toh, 2022; Chu et al., 2023; Yang et al., 2025). In our main result, i.e., Theorem 1, we keep Assumption 1 for the general fixed-step-size entropic scheme. Figure 2 shows the AAE of the iterates generated by (2) with $\alpha=0.5$, $\epsilon=1.0$ and various values of $\gamma$. This provides empirical evidence that the Assumption 1 holds. However, we also show that Assumption 1 is not ad hoc: in Appendix C we prove (c.f., Theorem 2) that, for the entropic generator, the accumulative asymmetric error is automatically summable under a mild "small enough" and decaying step-size regime ($\sum_{k=0}^\infty\gamma_k^3<\infty$). We are now ready to state our main result on the convergence of Eq. (2) to a first-order stationary point of (1).

**Theorem 1.** *Let $\mathcal{X}_1=(X_1,A_1,\mu)$, $\mathcal{X}_2=(X_2,A_2,\nu)$ be two featured measure networks, $C\in\mathbb{R}_+^{m\times n}$, $\alpha\in[0,1]$, $\epsilon>0$, with $C_\epsilon:=C-\epsilon 1_m 1_n^\top$, $\gamma>0$, and $T^{(0)}=\frac{1}{mn}1_m 1_n^\top$. Let Assumption 1 hold. Then, the iterates generated by (2) converge to a first-order stationary point of (1).*

## 2.2 NETWORK MOTIFS FOR ALIGNMENT OF ECOLOGICAL NETWORKS

Ecological networks are represented by directed graphs $\mathcal{G}=(X,\tilde{A})$, where $X$ is the set of species and $\tilde{A}$ is the adjacency matrix of the directed graph (not necessarily symmetric). We model ecological networks as featured measure networks $\mathcal{X}=(X,A,\mu,\mathfrak{m})$, where $A:=\tilde{A}+\tilde{A}^\top$ is the adjacency matrix of the underlying undirected graph of $\mathcal{G}$, $\mu$ is a measure reflecting the relative importance of species, and $\mathfrak{m}$ is the corresponding motif role profile for this network. Specifically, given $\mathcal{G}$, its *motif role profile* is a map $\mathfrak{m}_X:X\to\mathbb{R}^p$ defined as

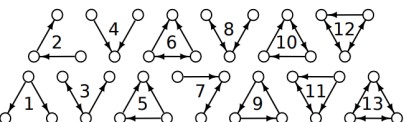

Figure 3: The 13 non-isomorphic three-node directed motifs (adapted from (Mora et al., 2018b)).

$$(\mathfrak{m}_X(x))_i=\#\{\text{occurrences of motif role } i \text{ that } x \text{ participates in } \mathcal{G}\},\quad\forall\,i=1,\ldots,p.$$

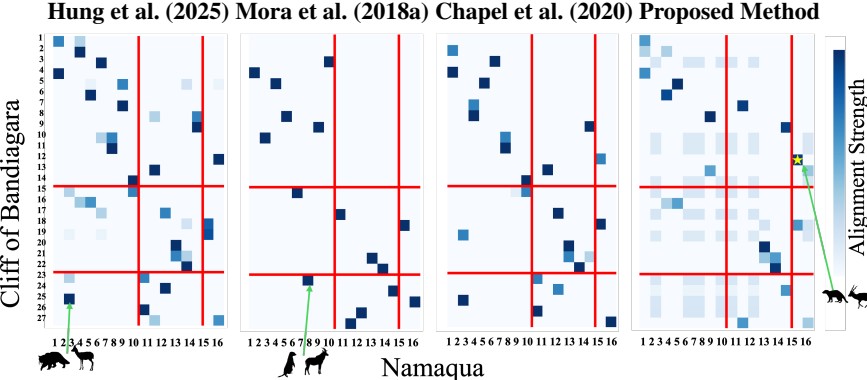

**Hung et al. (2025) Mora et al. (2018a) Chapel et al. (2020) Proposed Method**

Figure 5: Alignment results between Cliff of Bandiagara and Namaqua for various alignment methods: Red lines divide the species by dietary classification (carnivore, omnivore, herbivore), and species are sorted by decreasing biomass within each division. Number-species correspondences can be found in Appendix A.4.

The feature vector $\mathfrak{m}$ is defined for all directed unweighted networks; hence, its definition can carry over to arbitrary directed unweighted measure networks. Moreover, note that we decouple the directed graph into its (undirected) topological structure, captured by both $A$ and $\mathfrak{m}$, and the direction information, captured by $\mathfrak{m}$. In the rest of this paper, we will refer to this motif role profile map as $\mathfrak{m}$ when the domain of the map can be inferred. Figure 3 depicts the thirteen non-isomorphic three-node motifs for directed graphs, for a total of 30 unique role positions.

**Remark 2.1.** *We adopt the uniform probability distribution over species, i.e., $\mu = \nu_X/|X|$, i.e., all species are weighted equally. However, the formulation naturally allows for more general measures when abundance, biomass, or trait data are available.*

In (Mora et al., 2018a, Eq. (3)), the authors proposed the following cost to quantify the quality of a deterministic alignment between ecological networks $A$ and $B$:

$$e_{AB}(\Lambda) = \sum_{x \in \lambda} \left( \sum_{(\alpha,\beta) \in \Lambda_x} (1 - \rho(\alpha,\beta)) + \xi_x \right), \quad (3)$$

where $\Lambda$ is the set of pairs of aligned nodes, with $\alpha \in A$, and $\beta \in B$, $\Lambda_x \subseteq \Lambda$ is the set of pairs of nodes containing a neighbor $\alpha$ of $a$ and a neighbor $\beta$ of $b$, $\xi_x = (1-\epsilon) \max(k_{x_\alpha}, k_{x_\beta})$, where $k_{x_\alpha}$ (and $k_{x_\beta}$) is the number of neighbors of $a$ (and $b$) that are not paired with a neighbor of $a$ (and $b$), which can be understood as the penalty for the unpaired neighbors of every pairing $x = (a,b)$; finally, $\rho(\alpha,\beta)$ measures the species role similarity as the correlation coefficient of their respective network motifs. Moreover, the authors provide an annealing approach to compute alignments that achieve a low cost. However, in addition to only allowing deterministic alignments, results strongly depend on the stochasticity of the annealing process, and no guarantees of convergence to stationary points are provided. Our next theoretical result formalizes the heuristic developed in (Mora et al., 2018a) as a deterministic alignment computation, and shows its geometric structure as an optimization problem with a Gromov-Wasserstein-like cost functional.

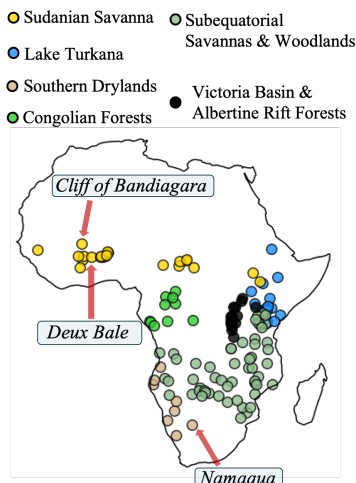

Figure 4: Locations of the 129 sites across Africa. The locations used in Fig. 5 are highlighted.

**Proposition 2.** *Let $\mathcal{X}_1 = (X_1, A_1, \nu_{X_1}, \mathfrak{m}_1)$ and $\mathcal{X}_2 = (X_2, A_2, \nu_{X_2}, \mathfrak{m}_2)$, with uniform measures, and $\alpha = 1$, $\epsilon > 0$. Then, a deterministic alignment minimizes (2) if and only if the generated alignment pairs minimize (3).*

## 3 NUMERICAL ANALYSIS ON SUB-SAHARAN MAMMAL FOOD WEBS

In this section, we demonstrate the interpretability, scalability, and versatility of the proposed framework. We use Python for the following experiments, using the NetworkX library by Hagberg et al.

(2008) for network data handling and the `pymfinder` package by Mora et al. (2018b) for network motif computations. Unless explicitly mentioned, we model all unweighted directed networks $\mathcal{G} = (X, \tilde{A})$ as the featured measure network $\mathcal{X} = (X, A, \nu_X, \mathfrak{m})$, where $A = \tilde{A} + \tilde{A}^\top$, $\nu_X$ is the uniform discrete measure over the vertex set $X$, and $\mathfrak{m}$ denotes the motif profile map.

We use a dataset of 129 Sub-Saharan African mammal food webs (Hung et al., 2025), which contains prey–predator interactions for large-bodied terrestrial mammals ($\geq 500$g). The dataset comprises 216 mammal species from 12 orders and 33 families. Figure 4 shows the locations of the food webs, classified by biomes. Interactions were compiled using a metaweb approach (Dunne, 2006), where a comprehensive binary interaction matrix was constructed from the union of predator–prey interactions from Mammals of Africa (Kingdon, 2013) and global mammal interaction data (Fricke et al., 2022), and additional inferred interactions based on taxonomic information and body size overlap (Brose et al., 2019; Gravel et al., 2013; Segar et al., 2020; Woodward et al., 2005). See Hung et al. (2025) for methodological details of the dataset assembly. Figure 6 provides a statistical overview of the dataset. Species richness across sites ranges from 16 to 75 and connectance from 0.0345 to 0.221 (Figs. 6a–6b). Taxonomically, the dataset spans herbivores($\approx 42.6\%$), omnivores($\approx 34.5\%$), and carnivores ($\approx 22.9\%$), with a detailed breakdown shown in Fig. 6c. While $14.8\%$ species are restricted to a single site, a large group recurs broadly across ecosystems (Fig. 6d).

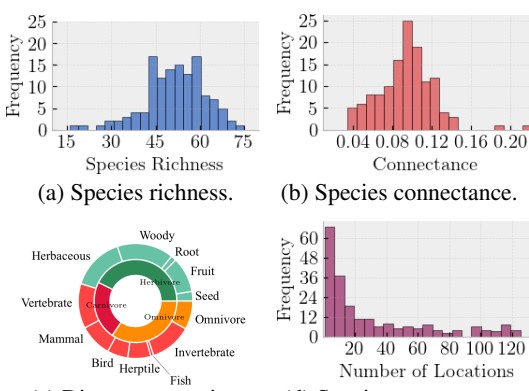

(a) Species richness.  (b) Species connectance.

(c) Dietary categories.  (d) Species occurrences.

Figure 6: Dataset Statistics

We first examine the food web alignments generated by various methods between the sites *Cliff of Bandiagara* and *Namaqua*. See Appendix A.5 for a visualization of these food webs. Notably, the Fused-Gromov Wasserstein(FGW) method by Hung et al. (2025) and the Partial-FGW method by Chapel et al. (2020) require additional feature information about the species. The feature dissimilarity required for those methods is computed by taking the Gower distance between the functional trait data for each pair of species. See Hung et al. (2025) for more information regarding its construction.

Figure 5 shows that the species within each dietary group are functionally similar to each other, with the exception of the alignment between *Mungos Mungo* (Banded mongoose, an insectivore) with *Raphicerus campestris* (Steenbok, a herbivore), marked with a star. Upon further inspection, we find that the prey of the banded mongoose all have body masses below $500g$ and are therefore excluded from the network. As a result, the banded mongoose does indeed play a functional role in *Cliff of Bandiagara* that is similar to that of the steenbok, testifying to the reliability of the proposed method. However, the exceptions found in the alignment produced by the

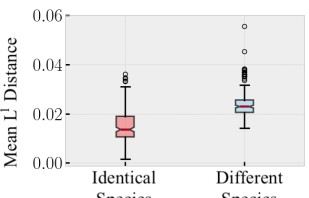

Figure 7: Alignment discrepancy of species.

FGW and Partial-FGW approaches are less ecologically consistent. For example, both of these methods aligned *Felis Silvestris* (Wildcat, a carnivore) to *Eudorcas rufifrons* (Red fronted gazelle, a herbivore). Unlike the prior case, the wildcat feeds on small mammals, four of which are also present in the *Namaqua* food web. On the contrary, the red fronted gazelle is near the bottom of the food chains in *Cliff of Bandiagara*, only very rarely feeding on *Ichneumia albicauda*, the white tailed mongoose. The wildcat and the gazelle play different functional roles in their ecological communities, but are nevertheless aligned strongly by the FGW and partial-FGW methods. The method in (Mora et al., 2018a) produces a similar exception where the *Hippotragus equinus* (Roan antelope, a herbivore) is aligned to the *Suricata suricatta* (Meerkat, a carnivore).

A challenge in the ecological network alignment task is to identify functionally overlapping or redundant species. Deterministic alignments produced by Mora et al. (2018a) overlook this phenomenon by aligning species with their single most similar counterparts. While an improvement from strictly deterministic alignments, we observe that the FGW and Partial-FGW alignment meth-

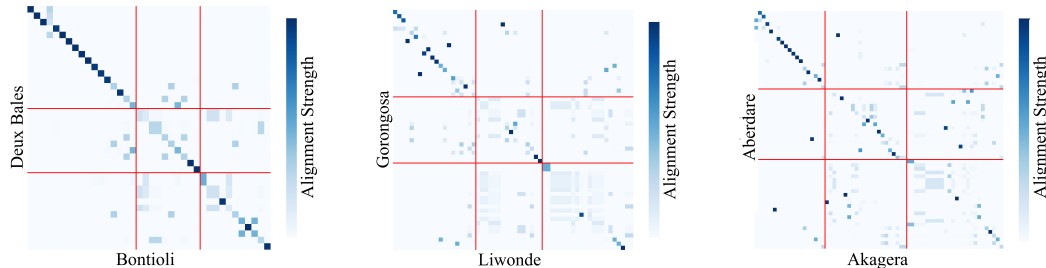

Figure 8: Additional non-deterministic alignments between sites. Red lines divide the species by dietary classification (carnivore, omnivore, herbivore), and species are sorted by biomass.

ods generate transport plans with as few alignments as possible. Figure 8 showcases three more alignments obtained using the proposed method. While the left alignment depicts two food webs with very few overlapping/redundant roles, the proposed alignments identify two groups of species that share very similar alignment distributions in the remaining two. This indicates that multiple species from the same network are functionally similar to the same set of species in the other network, suggesting that the functional roles of these species are also similar. In particular, these groups of species primarily fall within the omnivore and herbivore dietary classes (together accounting for about 75% of the entire dataset), suggesting that their functional roles are statistically more likely to overlap.

We performed network alignment using the proposed method pairwise across all locations in the dataset(with $\alpha = 0.5$ and $\epsilon = 1.0$), for a total of 8256 alignments. We perform a comprehensive hyperparameter analysis in Appendix A.7 to evaluate the sensitivity of our method to these choices. Although cost matrix construction is often the computational bottleneck ($\approx 0.15 \pm 0.07$ seconds of $0.20 \pm 0.08$ seconds per alignment), we could compute the motif embedding for each food web *a priori* and reduce the cost matrix computation to pairwise dissimilarity of the motif role profiles. This way, the computational bottleneck becomes the $\binom{N}{2}$ pairwise alignments, where $N$ is the dataset size. Given precomputed motif embeddings and running on an AMD Ryzen 7 9700X CPU, our method averaged $0.05 \pm 0.01$ seconds per alignment, which represents a 158x speedup compared to the 7.92 seconds average reported by the method in Mora et al. (2018a).

Figure 7 shows that our alignment method captures species' functional roles. In the absence of ground truth for ecological roles, we adopt the common assumption that identical species across different communities typically play more similar roles than arbitrary non-identical species. To test whether our alignments are consistent with this assumption, we defined the alignment discrepancy between two species as the average $L_1$ difference of their alignment distributions across networks. We then compared discrepancies between identical and non-identical species. Results show that identical species exhibited a mean discrepancy of $0.015 \pm 0.008$, significantly lower than the $0.024 \pm 0.007$ observed for non-identical species. These results demonstrate that the proposed method yields significantly more consistent alignments for identical species, verifying its ability to recover functional role similarities.

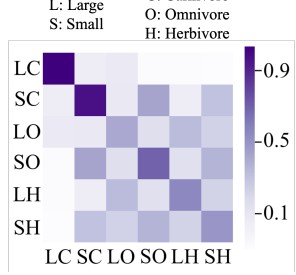

Figure 9: Weighted average of alignment strengths across and within groups of species.

Figure 9 shows the meta alignment obtained by partitioning the 216 species into six groups based on dietary class and median biomass. Each group is treated as a supernode, with alignment strengths computed as weighted averages of species-level alignments. For each pair of species and the alignment between them, the weight is defined as the product of the sizes of the two networks from which the two species are derived, so that the alignment strengths are comparable across networks. The results reveal strong alignments within large carnivores and within small carnivores, but a weak alignment between the two, consistent with the exclusion of small-prey species ($\leq 500g$) that lowers the trophic placement of small carnivores relative to larger ones.

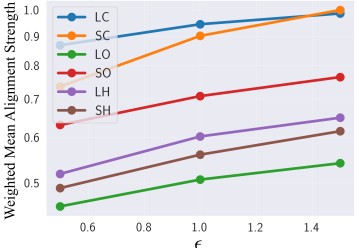

Figure 10: Meta alignment weights within each group of species vs $\epsilon$.

Figure 10 depicts within-group meta alignments across varying $\epsilon$. As $\epsilon$ increases, within-group strengths rise overall, with small carnivores showing the sharpest increase, surpassing large carnivores at $\epsilon = 1.5$. This pattern indicates that small carnivores are primarily self-aligned at low $\epsilon$ and only align with one another at higher values, highlighting their distinct functional roles. To further assess the robustness of our approach, we present results on alignments on additional real and synthetic data in Appendix A.8.

## 4 IDENTIFICATION OF NON-DETERMINISTIC BACKBONES OF INTERACTIONS

*Backbones of interactions* identify recurrent structural patterns in food webs that persist despite ecological or environmental variability. These backbones highlight trophic links that are consistently conserved across different ecosystems, thereby revealing ecological regularities that shape community assembly and constrain possible reorganization under disturbance. The existing approach to identifying backbones relies on deterministic alignments, where species are mapped one-to-one across networks (Mora et al., 2018a). However, the rigidity of deterministic methods is also evident in backbone identification. By recovering the full set of valid role correspondences within each pairwise alignment of ecological networks, non-deterministic alignments produce backbones that better reflect structural regularities across ecosystems. Formally, backbones are defined through role similarity scores that summarize how strongly a species aligns with counterparts across a dataset of ecological networks. Species with consistently high role similarity scores are considered central to the structural backbone, and the induced subgraph formed by these species provides a rigorous representation of the backbone of interactions. We now introduce the relevant definitions.

**Definition 4.1.** *Let $\mathcal{D} = \{\mathcal{X}_1, \ldots, \mathcal{X}_N\}$ denote a collection of featured measure networks, and let $\{T^{ij}\}_{i,j=1}^N$ be the pairwise non-deterministic alignments between them with corresponding dissimilarity matrices $\{C^{ij}\}_{i,j=1}^N$. For a species $u \in X_i$, its role similarity score $\mathcal{S}$ is defined as $\mathcal{S}(u) = \sum_{p=1}^N \sum_{v \in X_p} (1 - C_{uv}^{ip}) T_{uv}^{ip}$. A species achieves a high role similarity score when little of its mass is self-aligned and when it aligns with species that have a low dissimilarity cost across the dataset.*

**Definition 4.2** (Top-$k$ backbone). *Given a network $\mathcal{X}_i$ and the role similarity scores of its species, the top-$k$ backbone of $\mathcal{X}_i$, denoted $B_i = (V_{B_i}, E_{B_i})$, is the subgraph induced by the $k$ species in $X_i$ with the highest role similarity scores.*

We adopt the criteria of Mora et al. (2018a) to assess the internal consistency and ecological coherence of $B_i$: (i) *relative connectance*, the ratio of connectance within the backbone compared to the full network; (ii) *connectivity* ("path likelihood" in (Mora et al., 2018a)), the fraction of backbones that induce a connected subnetwork; and (iii) *transitivity*. While (i) and (ii) lift directly to our setting, (iii) requires a new definition because our proposed correspondences are fractional and many-to-many. Therefore, we introduce the following notion of non-deterministic transitivity:

**Definition 4.3** (Non-deterministic transitivity score). *For each species $u \in V_{B_i}$, its non-deterministic transitivity score is*

$$Tr(u) \triangleq \sum_{\substack{p,q \in [N] \\ p<q,\, p,q \neq i}} \sum_{v \in X_p} \sum_{w \in X_q} T_{uv}^{ip} \cdot T_{uw}^{iq} \cdot T_{vw}^{pq} \Big/ \sum_{\substack{p,q \in [N] \\ p<q,\, p,q \neq i}} \sum_{v \in X_p} \sum_{w \in X_q} T_{uv}^{ip} \cdot T_{uw}^{iq}.$$

*The transitivity score of a backbone is obtained by averaging $Tr(u)$ across all $u \in V_{B_i}$. It measures the consistency with which species alignments form transitive triangles across networks.*

We empirically analyze the backbones of interactions underlying the food webs in our dataset. Fig. 11 shows the top-7 backbones of two of our food webs. We first computed role similarity scores (Def. 4.1) for all species for each network, and sorted them in decreasing order. Then, for a given $k$, we compute the subnetwork induced by the top-$k$ species to obtain the top-$k$ backbone (Def. 4.2). We measure the connectivity and transitivity of our backbones to assess the quality of the generated backbones. In particular, for each backbone size $k$, we find the median relative connectance (Fig. 12a), path likelihood (Fig. 12b) and transitivity score (Fig. 12c, defined in Def. 4.3). These scores are compared to a null model, where the same computations are performed on the subnetwork induced by $k$ species that are randomly selected, averaged over 50 trials. Our backbones demonstrated a significantly higher relative connectance than the null model for smaller values of $k$. For large values of $k$, it is expected that the relative connectance of both our backbones and the null backbones converges to 1, because both backbones cover the vast majority of species in most

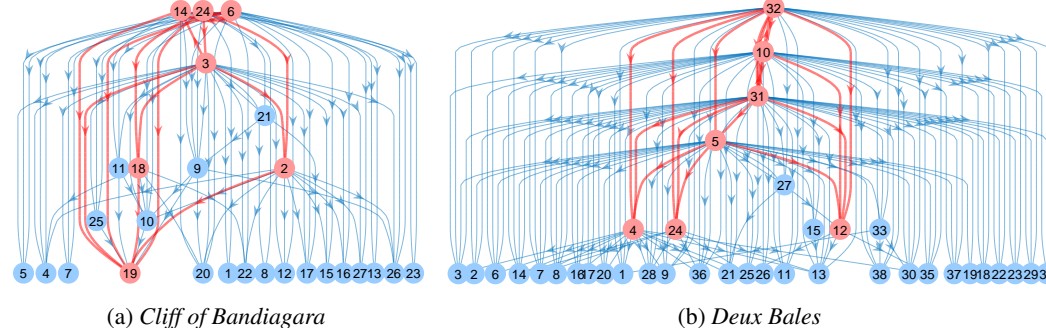

(a) *Cliff of Bandiagara*                    (b) *Deux Bales*

Figure 11: Examples of the top-7 backbones of food webs in our dataset. Species in the backbone are marked red, while the remaining species are blue. The full correspondence between number labels and species names can be found in Appendix A.4.

food webs, making them have similar connectance to the full web. Moreover, our backbones are much more likely to form a single connected component than the null model, as demonstrated by having more than half of the backbones connected with as few as 6 species, which is significantly fewer than the 19 species required for the null backbones. Lastly, our backbones are consistently more transitive than the null model, exhibiting high transitivity scores for all values of $k$ up to 7. Definition 4.2 involves summing over all triples of networks and all triples of backbone species, yielding time complexity $O(N^3 k^3)$, which is polynomial time and tractable for typical ecological datasets. In practice, the transitivity computation is dominated by the *null-model estimation*, which requires recomputing the metric across 50 random subsets. On our mid-range desktop CPU (AMD Ryzen 7 9700X), the entire null-model computation took approximately 20 hours, a cost incurred once solely for evaluation purposes. Overall, these findings demonstrate that our method produces backbones that consistently meet the backbone criteria proposed in Mora et al. (2018a), while also offering scalability and reproducibility.

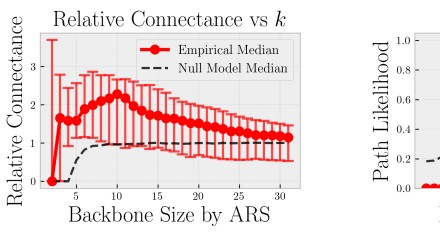
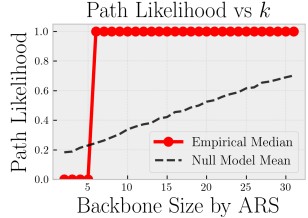
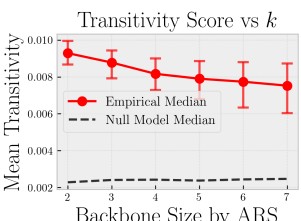

(a) Relative connectance, the ratio of backbone connectance to that of the full food web.

(b) Path likelihood, the fraction of backbones that induce a connected subnetwork.

(c) Transitivity, non-deterministic alignment transitivity score (Def. 4.3).

Figure 12: Structural properties of our top-$k$ backbones as a function of $k$.

## 5 CONCLUSIONS AND FUTURE WORK

We introduced *Wasserstein Motifs,* a non-deterministic network alignment framework for the ecological network alignment problem. We introduced a provably correct algorithm that efficiently computes species correspondences by functional role similarity. From these correspondences, we defined backbones of interactions underlying ecological networks. Numerical analysis performed on a continental-scale dataset of African mammal food webs showcased the capability of the proposed method to identify groups of functionally similar species, sidestepping species with redundant or overlapping roles. We evaluated the quality of the backbones derived from our alignments to discover that they demonstrate higher connectivity and transitivity than an appropriate null model.

Our experiments considered the uniform discrete measure over the species (Remark 2.1). Future work should restrict the measure to emphasize a focal subset of species, enabling zoomed-in analyses of functional role similarity. These extensions demonstrate the flexibility of the proposed framework in integrating heterogeneous ecological information while maintaining mathematical consistency. Moreover, parameter-free backbone constructions could be developed, such as backward backbones, where the least well-aligned species are removed in sequence until the scenario where removing the next species causes the food web to become disconnected.

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

# A APPENDIX

## A.1 MORE ALIGNMENT TOY EXAMPLES

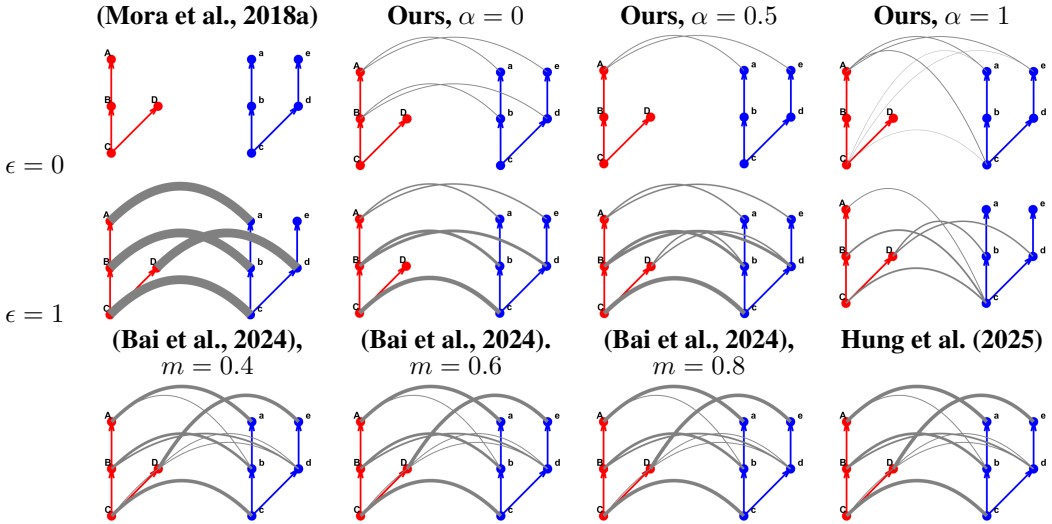

Figure 13: Optimal alignment between two toy networks with various alignment methods. Thicker lines indicate stronger alignment.

## A.2 TABLE OF COMPARISON

Table 1: Comparison with existing network alignment methods

| | DESIRED PROPERTY | | | |
|---|---|---|---|---|
| METHOD | NON-DET. | TOP.-AWARE | PARTIAL | MOTIF-BASED |
| Mora et al. (2018a) | ✗ | ✓ | ✓ | ✓ |
| Hung et al. (2025) | ✓ | ✓ | ✗ | ✗ |
| Bai et al. (2024) | ✓ | ✓ | ✓ | ✗ |
| Xu et al. (2019) | ✓ | ✓ | ✗ | ✗ |
| Ratnayaka et al. (2024) | ✗ | ✗ | ✓ | ✗ |
| Almulhim et al. (2019) | ✗ | ✓ | ✗ | ✓ |
| Wasserstein Motifs (Ours) | ✓ | ✓ | ✓ | ✓ |

A.3   ALGORITHM ANALYSIS AND DETAILS

We can rewrite our objective functional in Eq. (1) as follows:

$$g(\pi, w; C, \alpha, \epsilon) = \alpha \langle \pi, A_1(C \odot w)A_2 \rangle + (1-\alpha)\langle C, \frac{1}{2}(\pi + w) \rangle + \Xi_{\alpha,\epsilon}(\pi, w),$$

where $\Xi_{\alpha,\epsilon}(\pi, w) = -\epsilon(\alpha \langle \pi, A_1 w A_2 \rangle + \frac{1}{2}(1-\alpha)(\|\pi\|_1 + \|w\|_1))$, thus Eq. (1) is equivalent to

$$\min_{\pi=w} g(\pi, w; C, \alpha, \epsilon) + \mathbb{I}_{\mathcal{C}_1}(\pi) + \mathbb{I}_{\mathcal{C}_2}(w), \tag{4}$$

where $\mathbb{I}_{\mathcal{C}_i}$ denotes the indicator function for the constraint set $\mathcal{C}_i$. Then, we can penalize the equality constraint in the cost function and construct the penalized cost function:

$$\min_{\pi,w} g(\pi, w) + \mathbb{I}_{\mathcal{C}_1}(\pi) + \mathbb{I}_{\mathcal{C}_2}(w) + \frac{1}{\gamma} D_h(\pi, w), \text{ with } D_h(X, Y) \triangleq \sum_{ij} \left[ x_{ij} \log \frac{x_{ij}}{y_{ij}} \right],$$

where $D_h$ is the Bregman divergence induced by $h(X) = \sum_{ij} x_{ij} \log x_{ij}$, and $\gamma > 0$. Now, following Li et al. (2023), at iteration $k$, we alternate between linearizing $g$ around each operator ($\pi$ and $w$) and solve a KL–proximal step subproblem:

$$\pi^{k+1} = \arg\min_{\pi \in \mathcal{C}_1} \left\langle Q^{(k)}, \pi \right\rangle + \frac{1}{\gamma} D_h(\pi, w^k) \text{ and } w^{k+1} = \arg\min_{w \in \mathcal{C}_2} \left\langle Q^{'(k)}, w \right\rangle + \frac{1}{\gamma} D_h(w, \pi^{k+1}),$$

where the cost surrogates $Q^{(k)}$ and $Q^{'(k)}$ are defined as:

$$Q^{(k)} := \alpha A_1(C \odot w^k)A_2 + \frac{1}{2}(1-\alpha)C - \epsilon(A_1 w^k A_2 + \frac{1}{2}1_m 1_n^\top)$$

$$Q^{'(k)} := \alpha\, C \odot (A_1 \pi^{k+1} A_2) + \frac{1}{2}(1-\alpha)C - \epsilon(A_1 \pi^{k+1} A_2 + \frac{1}{2}1_m 1_n^\top).$$

In each of the subproblems, we minimize a linear term plus the Bregman divergence between the previous iteration of the other operator, under nonnegativity and a *single* set of marginal constraints.

We write the generic subproblem as $\min_{X \in \mathcal{C}} \langle Q, X \rangle + \frac{1}{\gamma} D_h(X, Y)$ with $\mathcal{C} \in \{\mathcal{C}_1, \mathcal{C}_2\}$. Setting $\mathcal{C} = \mathcal{C}_1$ and introducing dual variables $\lambda \in \mathbb{R}_+^m$, the Lagrangian is

$$\mathcal{L}(X, \lambda) := \langle Q, X \rangle + \frac{1}{\gamma} D_h(X, Y) + \langle \lambda, X1_n - \mu_1 \rangle \tag{5}$$

Minimizing w.r.t. $X$ entrywise yields, for any $x_{ij} > 0$:

$$0 = \frac{\partial \mathcal{L}}{\partial X_{ij}} = Q_{ij} + \frac{1}{\gamma} \log \frac{X_{ij}}{Y_{ij}} + \lambda_i \implies X_{ij} = Y_{ij} \exp\left( -\gamma(Q_{ij} + \lambda_i) \right) \tag{6}$$

If we denote $\hat{X}_{ij} = Y_{ij} \exp(-\gamma Q_{ij})$ as the unconstrained optimizer, then the Lagrangian optimizer can be written as $X_{ij}^\star = Y_{ij} \exp(-\gamma Q_{ij})\exp(-\gamma\lambda_i) = \hat{X}_{ij}\exp(-\gamma\lambda_i)$, with complementary slackness $\lambda_i^\star (X^\star 1_n - \mu_1)_i = 0$ for all $i \in [m]$. An analogous derivation with column constraints $\mathcal{C} = \mathcal{C}_2$ introduces $\eta \in \mathbb{R}_+^n$ and yields

$$X_{ij}^{'\star} = Y_{ij}' \exp(-\gamma Q_{ij}')\exp(-\gamma\eta_j^\star), \quad \eta_j^\star (X^{\star\top}1_n - \mu_2)_j = 0 \quad \forall j \in [n].$$

The structure of the Lagrangian optimizer shows that enforcing $\pi 1_n \preceq \mu_1$ (or $w^\top 1_m \preceq \mu_2$) is equivalent to scaling each row $i$ (or column $j$) of the unconstrained optimizer $\hat{X}$ by a factor $\exp(-\gamma\lambda_i)$ (or $\exp(-\gamma\eta_j)$). Complementary slackness implies that:

- if the current row sum $(\hat{X}1_n)_i \leq (\mu_1)_i$ (or column sum $(\hat{X}^\top 1_m)_j \leq (\mu_2)_j$), then $\lambda_i^* = 0$ (or $\eta_i^* = 0$) and we leave the row (or column) unchanged,
- if $(\hat{X}1_n)_i \geq (\mu_1)_i$ (or $(\hat{X}^\top 1_m)_j \geq (\mu_2)_j$), then $\lambda_i^* > 0$ (or $\eta_j^* > 0$) and the row (or column) is uniformly scaled until it satisfies the constraint.

Therefore, the Dykstra projection steps onto the constraint sets are defined as

$$P_{\mathcal{C}_1}(\hat{X}) := \text{diag}\left( \min\left( \frac{\mu_1}{\hat{X}1_n}, 1_m \right) \right) \hat{X}, \quad \text{and} \quad P_{\mathcal{C}_2}(\hat{X}) := \hat{X}\text{diag}\left( \min\left( \frac{\mu_2}{\hat{X}^\top 1_n}, 1_n \right) \right).$$

Algorithm 1 summarizes our algorithm.

---

**Algorithm 1:** KL-BAPG with Dykstra Projections

**Input :**
- featured measure networks $\mathcal{X}_1 = (X_1, A_1, \mu_1, \phi_1), \mathcal{X}_2 = (X_2, A_2, \mu_2, \phi_2)$.
- the feature discrepancy $d : \mathbb{R}^M \times \mathbb{R}^M \to \mathbb{R}_+$.
- the tradeoff parameter $\alpha \in [0, 1]$.
- the self-alignment regularization parameter $\epsilon > 0$.
- the step size $\gamma > 0$.
- stopping criteria for main loop (maximum iterations $K$, step tolerance, etc.).

**Output:** $T^\star$, an approximate solution to the non-deterministic alignment problem between $\mathcal{X}_1$ and $\mathcal{X}_2$.

**1** Normalize $\mu_1$ and $\mu_2$;

**2** $C_{ij} \leftarrow d(\phi_1(x^i), \phi_2(x^j)) - \epsilon$ for all $x^i \in X_1$ and $x^j \in X_2$ ;       `// Uniform shift`

**3** $T^{(0)} \leftarrow \frac{1}{mn} 1_m 1_n^\top$ ;       `// Initialize with uniform matrix`

**4 foreach** $k = 1, 2, \cdots, K$ **do**

**5**     $Q^{(k)} \leftarrow \alpha A_1 (C \odot T^{(k-1)}) A_2 + \frac{1}{2}(1 - \alpha)C$;

**6**     $\hat{T}^{(k)} \leftarrow T^{(k-1)} \odot \exp\left(-\gamma Q^{(k)}\right)$ ;       `// Unconstrained Lagrangian optimizer`

**7**     $\hat{T}^{(k)} \leftarrow \text{diag}(\min(\frac{\mu_1}{\hat{T}^{(k)} 1_n}, 1_m))\hat{T}^{(k)}$ ;       `// Proj. into row constraints`

**8**     $Q^{(k)'} \leftarrow \alpha C \odot (A_1 \hat{T}^{(k)} A_2) + \frac{1}{2}(1 - \alpha)C$;

**9**     $T^{(k)} \leftarrow \hat{T}^{(k)} \odot \exp(-\gamma Q^{(k)'})$ ;       `// Unconstrained Lagrangian optimizer`

**10**    $T^{(k)} \leftarrow T^{(k)}\text{diag}(\min(\frac{\mu_2}{T^{(k)\top} 1_m}, 1_n))$ ;       `// Proj. into column constraints`

**11**    Break if any stopping criterion is met;

**12 return** $T^{(k)}$;

---

## A.4 FOOD WEB SPECIES CORRESPONDENCES

Table 2: Species correspondences in *Namaqua* site

| Index | Species Name | Common Name |
|-------|--------------|-------------|
| 1 | Caracal_caracal | Caracal |
| 2 | Proteles_cristata | Aardwolf |
| 3 | Felis_silvestris | Wildcat |
| 4 | Vulpes_chama | Cape fox |
| 5 | Ictonyx_striatus | Striped polecat |
| 6 | Cynictis_penicillata | Yellow mongoose |
| 7 | Herpestes_pulverulentus | Cape gray mongoose |
| 8 | Suricata_suricatta | Meerkat |
| 9 | Sylvicapra_grimmia | Common duiker |
| 10 | Hystrix_africaeaustralis | Cape porcupine |
| 11 | Otocyon_megalotis | Bat eared fox |
| 12 | Lepus_capensis | Cape hare |
| 13 | Lepus_saxatilis | Scrub hare |
| 14 | Genetta_genetta | Common genet |
| 15 | Raphicerus_campestris | Steenbok |
| 16 | Procavia_capensis | Rock hyrax |

Table 3: Species correspondences in *Cliff of Bandiagara* site.

| Index | Species Name | Common Name |
|-------|--------------|-------------|
| 1 | Crocuta_crocuta | Spotted hyena |
| 2 | Panthera_pardus | Leopard |
| 3 | Aonyx_capensis | African clawless otter |
| 4 | Caracal_caracal | Caracal |
| 5 | Leptailurus_serval | Serval |
| 6 | Canis_adustus | Side striped jackal |
| 7 | Mellivora_capensis | Honey badger |
| 8 | Felis_silvestris | Wildcat |
| 9 | Herpestes_ichneumon | Egyptian mongoose |
| 10 | Vulpes_pallida | Pale fox |
| 11 | Ictonyx_libycus | Saharan striped polecat |
| 12 | Mungos_mungo | Banded mongoose |
| 13 | Ictonyx_striatus | Striped polecat |
| 14 | Herpestes_sanguineus | Common slender mongoose |
| 15 | Orycteropus_afer | Aardvark |
| 16 | Hyaena_hyaena | Striped hyena |
| 17 | Civettictis_civetta | African civet |
| 18 | Chlorocebus_sabaeus | Green monkey |
| 19 | Ichneumia_albicauda | White tailed mongoose |
| 20 | Lepus_capensis | Cape hare |
| 21 | Genetta_genetta | Common genet |
| 22 | Lepus_victoriae | African savanna hare |
| 23 | Hippotragus_equinus | Roan antelope |
| 24 | Papio_anubis | Olive baboon |
| 25 | Eudorcas_rufifrons | Red fronted gazelle |
| 26 | Erythrocebus_patas | Common patas monkey |
| 27 | Procavia_capensis | Rock hyrax |

## A.5 FOOD WEB VISUALIZATION

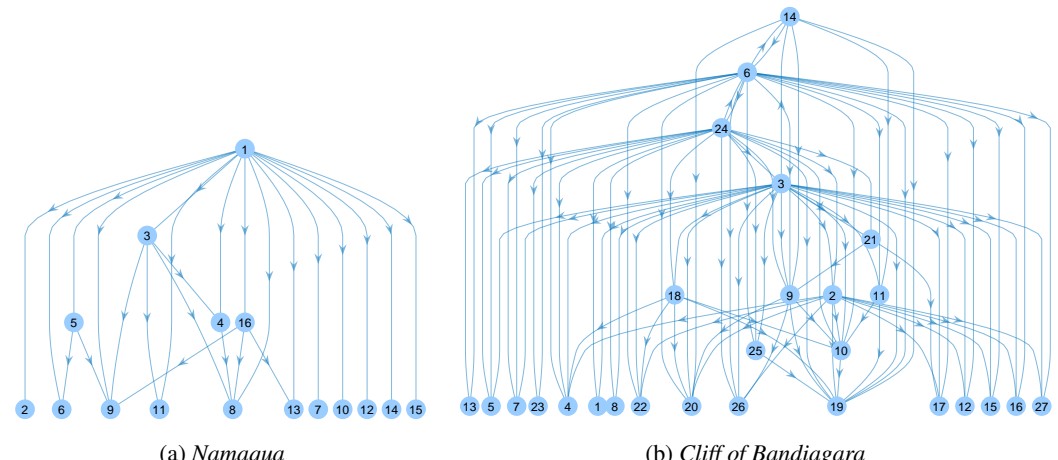

(a) *Namaqua*  (b) *Cliff of Bandiagara*

Figure 14: Examples of food webs as directed graphs.

## A.6 MOTIF-BASED FUSED GROMOV-WASSERSTEIN VARIANTS

**First-order alignment discrepancy vs. Gromov-Wasserstein**   We further expound on the difference between our proposed formulation 1 and that of Fused Gromov-Wasserstein (FGW). We emphasize the difference between our first-order term and its analogue in FGW (i.e., the Gromov-Wasserstein term).

Given a pair of networks $G_1$ and $G_2$, our first-order term is

$$\langle T, A_1(T \odot C)A_2 \rangle = \sum_{i,j,i',j'} (A_1)_{ii'}(A_2)_{jj'}C_{i'j'} T_{ij}T_{i'j'},$$

whereas the GW distance term is

$$\sum_{i,j,i',j'} L(d_1(x_i, x_{i'}), d_2(y_j, y_{j'})) T_{ij}T_{i'j'}.$$

The critical difference between these terms lies in our cost matrix $C$, which is comprised of distances between the features of a pair of nodes $v'_i \in G_1$ and $v'_j \in G_2$. The GW distance, on the other hand, assumes that the spaces it is computing the distance between are incomparable Mémoli (2011), implying that a cost between $G_1$ and $G_2$ cannot appear. Therefore, there is no choice of $L$ for which our first-order term and the GW distance are equal, in general. In the case that $L(x, y) = xy$ and $C$ is the all ones matrix, these terms coincide, but this case is not practical as it implies that all nodes have the same motif profile.

**Baseline Comparison.**   In order to determine whether the improved performance of our method over Gromov-Wasserstein based methods is due to the formulation of the cost function or the choice of features, we modify the existing baselines by incorporating the motif-role dissimilarity. In particular, we consider the following baselines, the last two of which are augmented with motif-based features/distances:

- **Vanilla FGW** Hung et al. (2025): Gower trait distance; shortest-path distance as metric (shown in Figs. 16 and 17).
- **FGW (traits as features, motifs as metric):** Gower trait distance as features; motif-role dissimilarity as metric (shown in Figs. 18 and 19).
- **FGW (motifs as features and metric):** motif-role dissimilarity used for both features and metric (shown in Figs. 20 and 21).

All baselines were assessed against the $L_1$ alignment discrepancy (Fig. 7), meta-web alignment (Fig. 9), and (3) backbone quality (Fig. 12) produced by our method.

All FGW variants separate identical vs. non-identical species in the discrepancy test. For meta-alignment, FGW with trait features performs well, as biomass and diet directly encode functional roles. The motif-only FGW variant performs noticeably worse for large carnivores, demonstrating that simply inserting motif features into FGW does not reproduce the expressivity of our structural term. FGW-based backbones also exhibit structural inconsistencies:

- **Vanilla FGW:** connected but **non-transitive backbones**,
- **FGW (traits as features, motifs as metric):** transitive but **largely disconnected**,
- **FGW (motifs as both features and metric): lower connectance** than Wasserstein Motifs.

Our method, in contrast, consistently produces backbones that are both connected and transitive, uniquely allowing *controllable self-alignment* via the regularizer $\epsilon$, a capability that FGW lacks. Overall, these findings demonstrate that our improvements stem from the *objective design*, not from feature choice, and that FGW/Partial-FGW baselines are neither structurally nor computationally comparable to our method.

**On Partial-FGW.**   We additionally examined Partial-FGW, as it is the closest FGW-based approach conceptually aligned with our goal of non-deterministic correspondences. However, we found it to be computationally infeasible at ecological scales. In an experiment computing pairwise

alignments across a representative subset of networks (spanning the full range of 20–75 species), Partial-FGW required *up to 15 seconds per alignment* for networks with ∼ 60 nodes (as shown in Fig. 15). In contrast, FGW and our method required about ∼0.05 seconds. Since our analysis requires 8256 pairwise alignments, Partial-FGW would require several days of computation and is therefore not a practical baseline for ecological network data. This is consistent with prior reports of Partial-FGW scaling poorly when structural consistency dominates the cost.

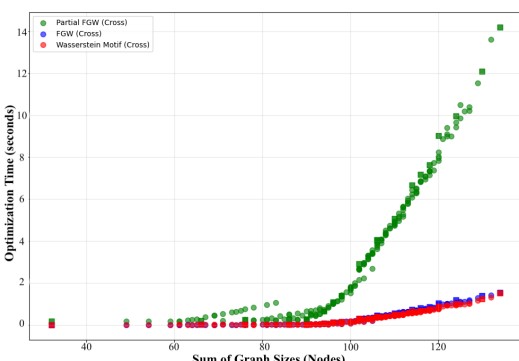

Figure 15: Optimization Time vs Graph Sizes for our method and the Baselines.

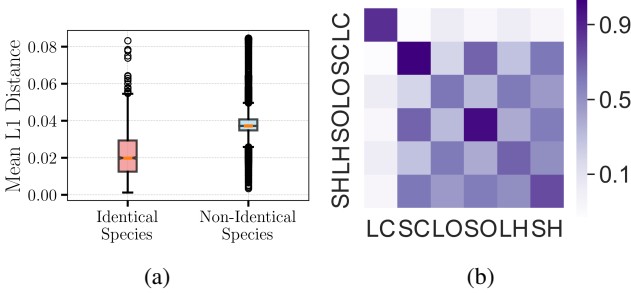

Figure 16: Alignment Discrepancy and Meta-alignment for Vanilla FGW

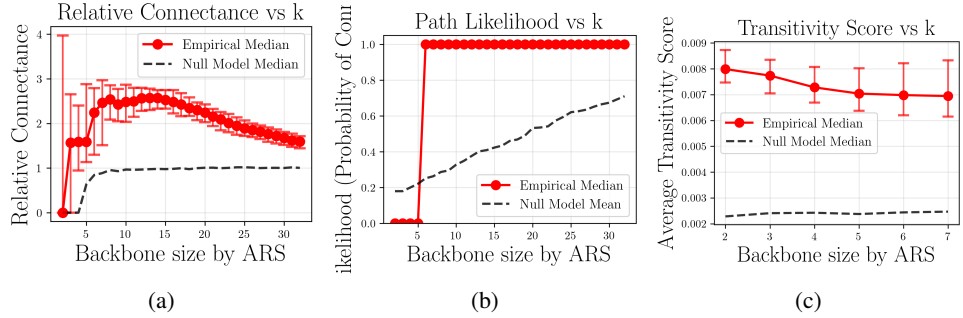

(a)                     (b)                     (c)

Figure 17: Statistical Analysis of Top-k backbones for Vanilla FGW

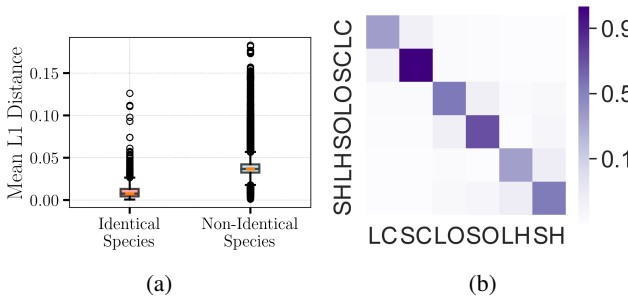

(a)                    (b)

Figure 18: Alignment Discrepancy and Meta-alignment for FGW with Traits as features and motif-role dissimilarity as metric

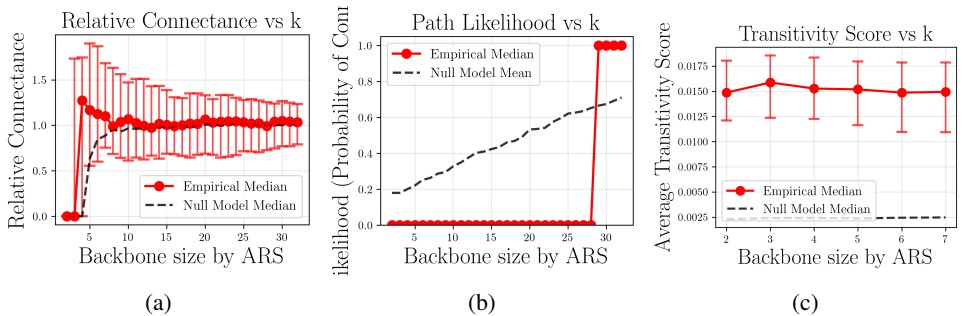

(a)                    (b)                    (c)

Figure 19: Statistical Analysis of Top-k backbones for FGW with Traits as features and motif-role dissimilarity as metric

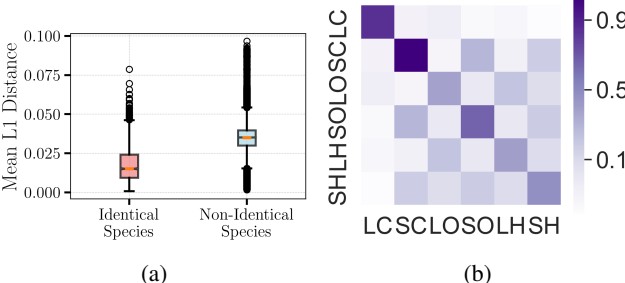

(a)                    (b)

Figure 20: Alignment Discrepancy and Meta-alignment for FGW with motif-role based features and motif-role dissimilarity as metric

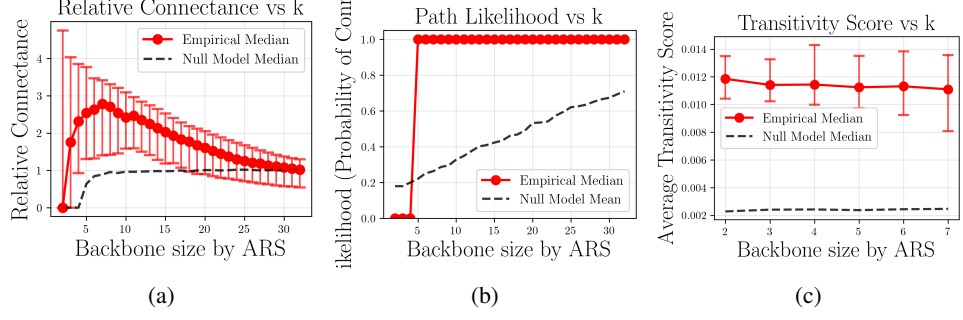

(a)                    (b)                    (c)

Figure 21: Statistical Analysis of Top-k backbones for FGW with motif-role based features and motif-role dissimilarity as metric

### A.7 HYPERPARAMETER ANALYSIS

**Sensitivity Analysis.** In this section, we conduct sensitivity analyses on the model parameters $(\alpha, \epsilon, \gamma)$, and clarify how they interact.

**The tradeoff parameter $\alpha$.** We set $\alpha = 0.5$ as the default to reflect an equal balance between the zeroth-order (feature-driven) and first-order (structure-driven) terms in Equation (1). This choice corresponds to the natural modelling assumption that motif-role similarity and structural similarity contribute comparably to ecological functional equivalence. In Figure 22(a), we found that both the zeroth-order and first-order cost alignment costs are near-constant, and that the first-order cost is larger than the zeroth-order. This justifies $\alpha = 0.5$ as a reasonable trade-off between the respective costs. Moreover, in Figure 22(b), we found that the solution varies smoothly with $\alpha$: perturbing $\alpha$ by $\pm 0.1$ changes the alignment by only $\sim 0.12$ in Frobenius norm. This small and approximately linear response demonstrates that (i) $\alpha = 0.5$ is a robust and interpretable default, and (ii) the method is not sensitive to moderate deviations from this value.

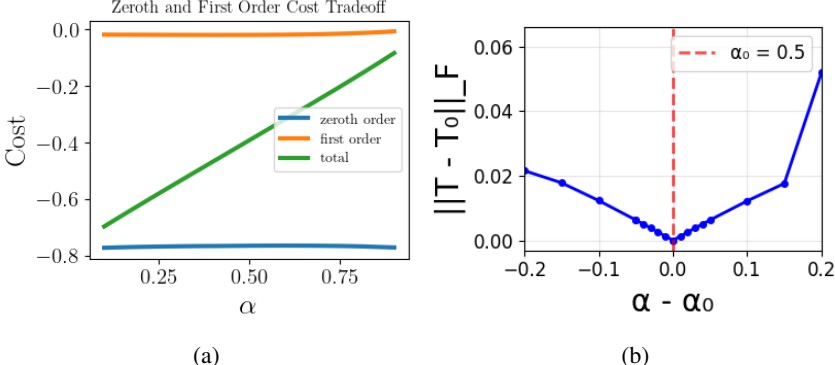

(a)                                    (b)

Figure 22: $\alpha$ Sensitivity

**The self-alignment regularizer $\epsilon$.** We additionally examined how the total transported mass (controlled directly by $\epsilon$) varies with $\epsilon$. For a pair of food webs with sizes $m$ and $n$, respectively, the relationship follows a smooth saturating curve well-approximated by

$$f(\epsilon) = a\big(1 - \exp(-b\epsilon)\big),$$

where $a \approx 0.83$ and $b \approx 4.25(m + n) - 56.5$. Figure 23 shows this approximation on three pairs of real food webs, with a mean $R^2$ value of $0.984$. This indicates that $\epsilon$ influences the degree of self-alignment in a *stable and predictable* manner. This also yields a practical scheme for selecting $\epsilon$ given a desired level of tolerated self-alignment.

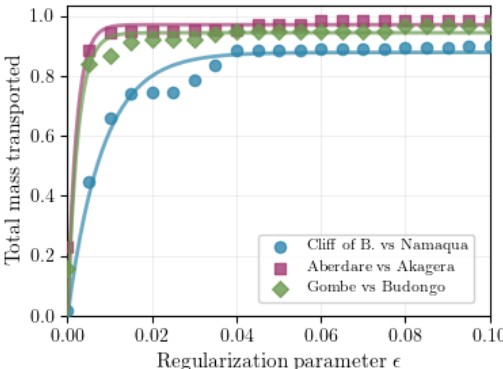

Figure 23: Approximated and empirical total mass transported vs. $\epsilon$ for three pairs of food webs.

**The step size $\gamma$.** In the problem formulation and algorithm, $\gamma$ effectively plays the role of a step-size in the usual sense of a first-order optimization algorithm. We found $\gamma = 0.1$ to be a suitable default value that strikes a balance between stability and the speed of convergence. To test this, we align several pairs of networks, using different values of $\gamma$, and show robustness to the choice of this parameter. Our theory shows convergence for any $\gamma > 0$ as shown in Theorem 1, thus the choice of $\gamma$ controls convergence rates, not convergence properties. In Fig. 24 we show that, in producing an alignment between *Cliff of Bandiagara* and *Namaqua*, our algorithm smoothly minimizes the cost with a $\gamma$-depdendent rate of convergence.

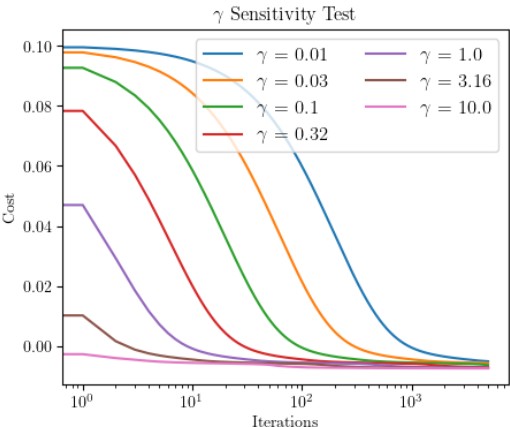

Figure 24: $\gamma$ sensitivity

**Ablation Analysis.** To further clarify the role of individual terms in the objective, we conducted two ablations:

- **Self-alignment removed ($\epsilon = 0$):** Performance deteriorates sharply. Meta-alignment consistency collapses, and backbones fall below the null model in transitivity, confirming that the self-alignment penalty is essential to avoid trivial low-mass solutions (shown in Figs. 25, 26).

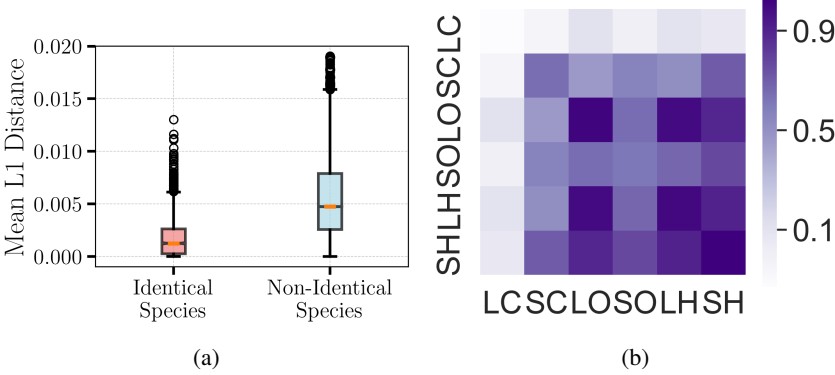

(a)                                       (b)

Figure 25: Alignment Discrepancy and Meta-alignment for $\epsilon = 0$

- **First-order term removed ($\alpha = 0$):** Results remain comparable on most tasks, showing that the model is *robust* to the tradeoff parameter. Since $\alpha$ interpolates between two ecologically meaningful similarity sources (motif-based features and structural interactions), $\alpha = 0$ represents a valid special case rather than a failure mode (shown in Figs. 27, 28).

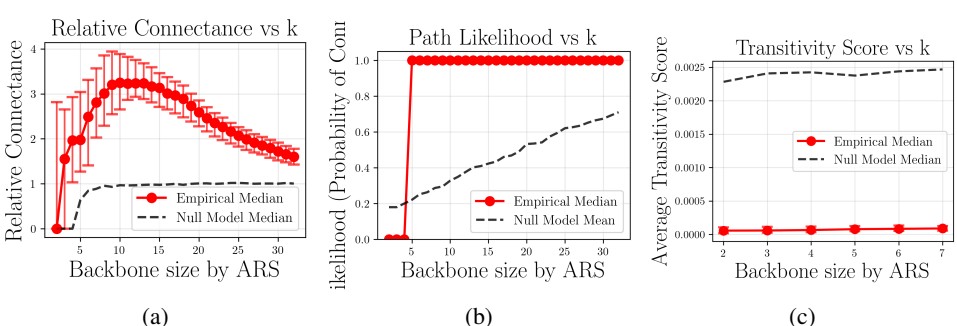

Figure 26: Statistical Analysis of Top-k backbones for $\epsilon = 0$.

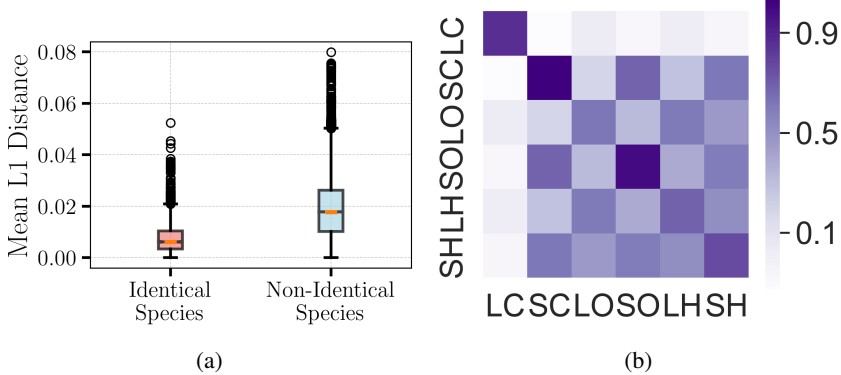

Figure 27: Alignment Discrepancy and Meta-alignment for $\alpha = 0$

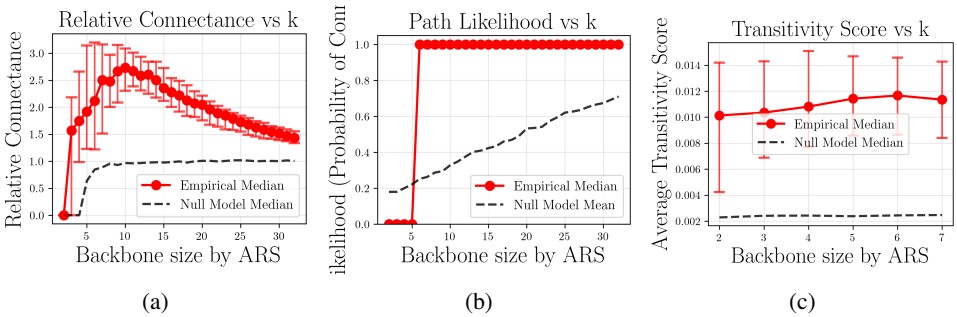

Figure 28: Statistical Analysis of Top-k backbones for $\alpha = 0$

## A.8 ALIGNMENTS ON ADDITIONAL REAL AND SYNTHETIC DATA

To demonstrate broader applicability to ecological networks, we also evaluated our method on a subset of food webs from Mora et al. (2018a), for which we were able to obtain the directed interaction data. We performed pairwise alignments between selected pairs of food webs, shown in Figure 29. However, the dataset does not include species-level trait profiles or other ecological attributes that would allow for quantitative ecological validation (e.g., comparing aligned species by diet, body size, or functional role). As a result, our evaluation here is restricted to qualitative inspection of alignment structure rather than trait-based assessment.

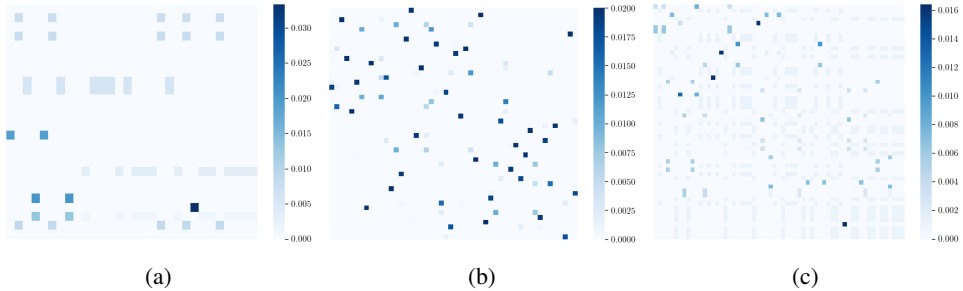

(a)                          (b)                          (c)

Figure 29: Examples of pairwise alignments between the dataset used in Mora et al. (2018a). (Left) Coachella (26 species) vs. Connery (30 species); (Middle) Mondego (48 species) vs. Reef (50 species); (Right) Alford (56 species) vs. Beaver (61 species).

To further demonstrate general utility, we additionally aligned synthetic ecological networks generated by the classical Niche model (Williams & Martinez, 2000) and Cascade model (Cohen & Newman, 1985). (Fig. 30, Fig. 31) In both the Niche-model and Cascade-model experiments, we observe that species with similar niche values, which correspond to similar trophic positions in the generative models, tend to align strongly with one another. This is exactly the structural pattern predicted by these models: species with comparable niche parameters have overlapping feeding ranges and play analogous roles in the synthetic food web. The fact that our method reconstructs these relationships from network structure alone, without access to the underlying niche values, demonstrates that the alignment objective captures meaningful functional similarities. This serves as an additional validation step, showing that Wasserstein Motifs recovers ecologically coherent correspondences even in controlled synthetic systems where the ground-truth generative mechanism is known.

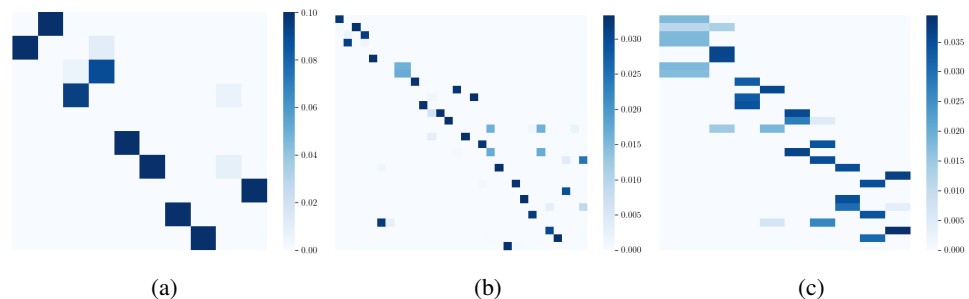

(a)                          (b)                          (c)

Figure 30: Examples of pairwise alignments between synthetic food webs generated by the Niche model (Williams & Martinez, 2000). Species are sorted in decreasing niche values, which represent trophic levels. (Left) 10 species, expected connectance 0.4 vs. 10 species, expected connectance 0.3; (Middle) 30 species, expected connectance 0.3 vs. 30 species, expected connectance 0.25; (Right) 30 species, expected connectance 0.3 vs. 10 species, expected connectance 0.4.

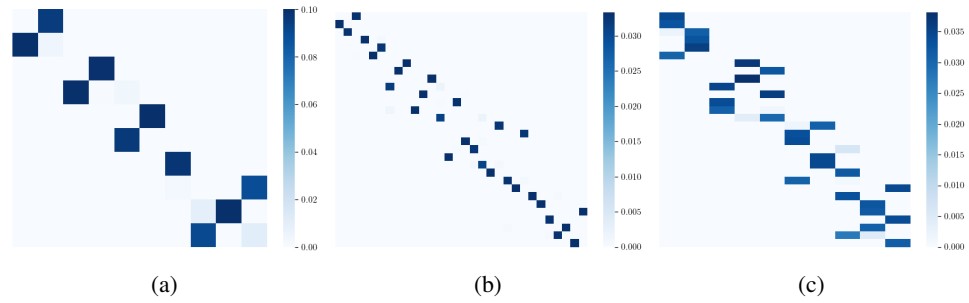

          (a)                         (b)                        (c)

Figure 31: Examples of pairwise alignments between synthetic food webs generated by the Cascade model (Cohen & Newman, 1985). Species are sorted in decreasing niche values, which represent trophic levels. (Left) 10 species, expected connectance 0.4 vs. 10 species, expected connectance 0.3; (Middle) 30 species, expected connectance 0.3 vs. 30 species, expected connectance 0.25; (Right) 30 species, expected connectance 0.3 vs. 10 species, expected connectance 0.4.

## A.9 Wasserstein Motifs Pipeline

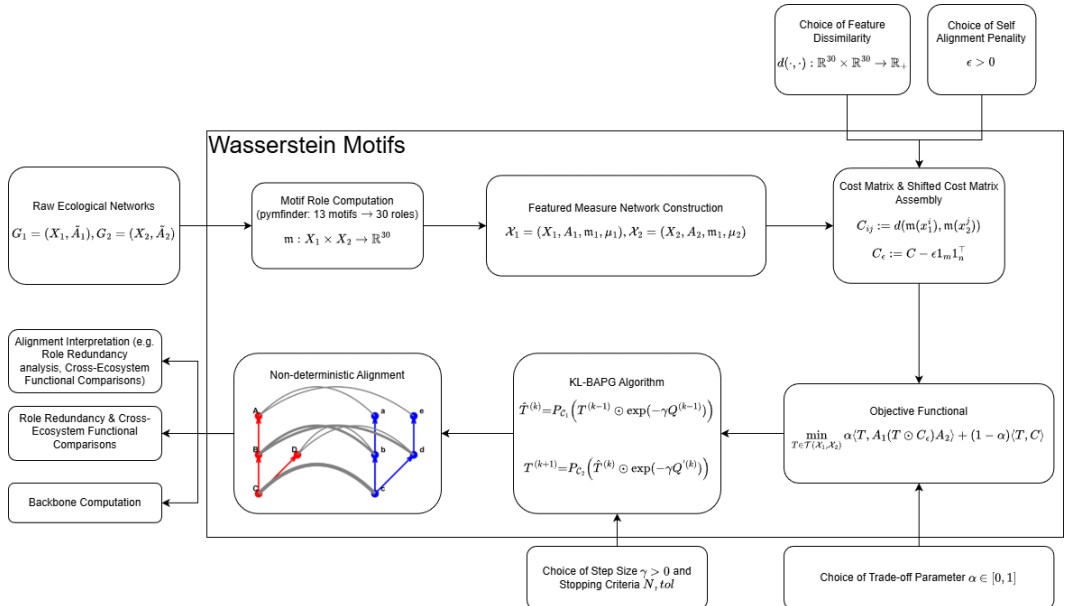

Figure 32: Pipeline Diagram for Wasserstein Motifs Framework.

## A.10 DIRECTED ADJACENCY VARIANT

This section explores the variant of Wasserstein Motifs where directed adjacency is used instead of its undirected counterpart in Equation 1. Importantly, none of our theoretical results require the adjacency matrices $A_1$ and $A_2$ to be symmetric. The first-order term

$$\langle T, \ A_1(T \odot C)A_2 \rangle$$

remains well-defined for any nonnegative matrices, and directed adjacencies (e.g., separate in-/out-adjacency matrices) are therefore a drop-in replacement in our framework.

However, using directed adjacencies changes the ecological interpretation of the first-order similarity. In this variant, the structural term compares only *directed neighbors* (e.g., prey sets), whereas our main formulation averages over predator–prey interactions by symmetrizing the adjacency. This shift places stronger weight on trophic directionality and weaker weight on undirected neighborhood structure.

To evaluate this directed variant, we repeated all key experiments - pairwise alignments, role-consistency analysis, meta-alignment, and backbone quality - with directed adjacency matrices. Figure 33 and 34 report the results. While the method remains capable of distinguishing identical species from non-identical ones, we observed two consistent degradations:

1. **Meta-alignment performance declined substantially.** Alignment strengths across networks became noisier, and trophic-group structure was less pronounced.

2. **Backbone quality weakened.** Although the resulting top-$k$ backbones remained connected, their non-deterministic transitivity was only slightly above the null model, which is far below the improvement observed under our undirected formulation.

Overall, while the framework technically supports directed adjacencies, our empirical analysis indicates that the directed version yields weaker alignment coherence and backbone structure. We attribute this degradation to the loss of structural information when only outgoing interactions (prey) are considered, whereas ecological role similarity is typically driven jointly by both predator and prey relationships.

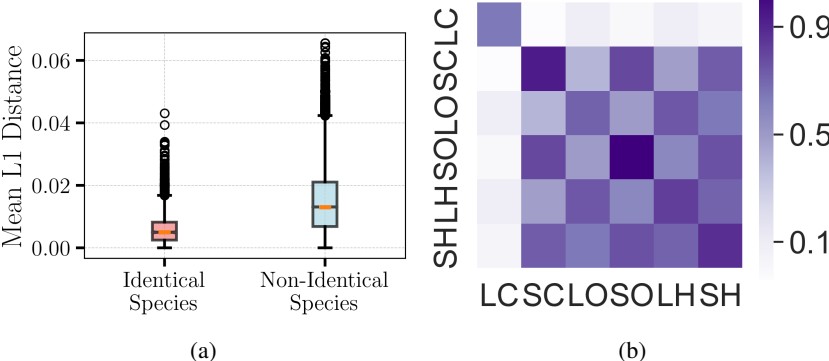

(a)                 (b)

Figure 33: Alignment Discrepancy and Meta-alignment for the Directed-adjacency Variant

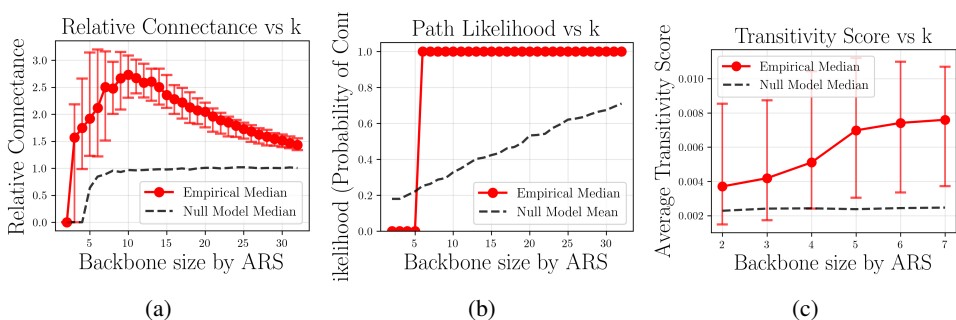

Figure 34: Statistical Analysis of Top-k backbones for the Directed-adjacency Variant

# B PROOF OF CLAIMS

## B.1 PROOF OF THEOREM 1

We first write the split form of the problem as seen in Equation (4):

$$\min_{\pi=w} g(\pi, w) + \mathbb{I}_{\mathcal{C}_1}(\pi) + \mathbb{I}_{\mathcal{C}_2}(w),$$
$$\text{where } \mathcal{C}_1 = \{X \in [0,1]^{m \times n} : X1 \preceq \mu\}, \quad \mathcal{C}_2 = \{X \in [0,1]^{m \times n} : X^\top 1 \preceq \nu\}. \tag{7}$$

Let $h(X) = \sum_{ij} x_{ij} \log x_{ij}$ be the negative entropy function and $D_h$ be its Bregman divergence. Let $\{(\pi^{(k)}, w^{(k)})\}_{k \geq 0}$ be the sequence generated by alternatively solving the subproblems in (2).

We define the potential function:

$$F_\gamma(\pi, w) \triangleq g(\pi, w) + \mathbb{I}_{\mathcal{C}_1}(\pi) + \mathbb{I}_{\mathcal{C}_2}(w) + \frac{1}{\gamma} D_h(\pi, w).$$

**Lemma 1.** *The following properties hold:*

1. $f(\pi, w) = \alpha \langle \pi, A_1(C \odot w)A_2 \rangle$ *is bilinear;*

2. $\mathcal{C}_1$ *and* $\mathcal{C}_2$ *are closed, convex polytopes;*

3. *the potential function* $F_\gamma$ *is coercive.*

*Proof of Lemma 1.* The three statements are proved in parallel:

1. This follows immediately due to the linearity of inner products, matrix multiplication, and the Hadamard product.

2. **Closedness:** Both inequality constraints $X1 \preceq \mu$ (resp. $X^\top 1 \preceq \nu$) can be represented as an intersection of closed half-spaces. Since $[0,1]^{m \times n}$ is a product of closed intervals which is also closed, we have that the intersection of closed sets is closed; hence, $\mathcal{C}_1$ and $\mathcal{C}_2$ are closed.
   **Convexity:** Both the unit box $[0,1]^{m \times n}$ and the linear inequalities are convex, and hence their intersection is convex.
   **Polytope:** Both $\mathcal{C}_1$ and $\mathcal{C}_2$ are bounded by the unit box $[0,1]^{m \times n}$. In addition, they are both given by finitely many linear inequalities, so they are polyhedra. Since they are both bounded polyhedra, by definition, they are polytopes.

3. Since both $\mathcal{C}_1$ and $\mathcal{C}_2$ are bounded, $\mathcal{C}_1 \times \mathcal{C}_2$ is bounded. Hence, whenever $\|(\pi, w)\| \to \infty$, we must have that $(\pi, w) \notin \mathcal{C}_1 \times \mathcal{C}_2$, which implies either $\pi \notin \mathcal{C}_1$ or $w \notin \mathcal{C}_2$. In the former case, we have that $\mathbb{I}_{\mathcal{C}_1}(\pi) \to \infty$, and in the latter case we have that $\mathbb{I}_{\mathcal{C}_2}(w) \to \infty$, both of which implies $F_\gamma(\pi, w) \to \infty$, as desired.

$\square$

*Proof of Theorem 1.* Since the accumulative asymmetrical error is bounded and the statements in Lemma 1 hold, then by (Li et al., 2023, Theorem 3.6), every limit point of the sequence $\{(\pi^{(k)}, w^{(k)})\}_{k \geq 0}$ generated by (2) belongs to the fixed point set of the Bregman Alternating Projected Gradient. Hence, the updates (2) generate a sequence $\{T^{(k)}\}_{\geq 0}$ that converges to a first-order stationary point of Problem 7, and therefore a first-order stationary point of Equation (1). $\square$

## B.2 PROOF OF PROPOSITION 1

*Proof of Proposition 1.* Let $\mathcal{X}_1 = (X_1, A_1, \nu_{X_1})$ and $\mathcal{X}_2 = (X_2, A_2, \nu_{X_2})$ be measure networks with the uniform discrete measures over their vertex sets. Recall from Equation (1) that the feasible set for deterministic alignment is

$$\left\{ T \in \{0,1\}^{m \times n} : T1_n \preceq \nu_{X_1}, T^\top 1_m \preceq \nu_{X_2} \right\}.$$

Since $\nu_{X_1}$ and $\nu_{X_2}$ each assign one unit of mass to each vertex, the marginal constraints can be further reduced to

$$\sum_{j=1}^{n} T_{ij} \leq 1 \quad \text{and} \quad \sum_{i=1}^{m} T_{ij} \leq 1. \tag{8}$$

Consider the complete bipartite graph $K_{m,n} = (X_1 \cup X_2, E)$ with partition $(X_1, X_2)$. A (not necessarily perfect) matching $\mathcal{M} \subseteq E$ is a set of vertex-disjoint edges.

We now construct an explicit bijection between matchings of $K_{m,n}$ and feasible deterministic alignments.

**Matchings $\rightarrow$ Deterministic Alignments.** Given a matching $\mathcal{M}$, define $T^{\mathcal{M}} \in \{0,1\}^{m \times n}$ by

$$T_{ij}^{\mathcal{M}} = \begin{cases} 1 & \text{if } (x_i, x_j') \in \mathcal{M}, \\ 0 & \text{otherwise.} \end{cases}$$

Since $\mathcal{M}$ is a matching, no vertex is incident to more than one edge. Hence $T^{\mathcal{M}}$ satisfies (8) and is a deterministic alignment.

**Deterministic Alignments $\rightarrow$ Matchings.** Conversely, let $T$ be a deterministic alignment. Define

$$\mathcal{M}^T = \{(x_i, x_j' \in E : T_{ij} = 1)\}.$$

The marginal constraints in (8) imply that each $x_i \in X_1$ and $x_j' \in X_2$ is incident to at most one edge in $\mathcal{M}^T$, making $\mathcal{M}^T$ a valid matching.

**Bijection.** Clearly,

$$\mathcal{M}^{T^{\mathcal{M}}} = \mathcal{M} \quad \text{and} \quad T^{\mathcal{M}^T} = T,$$

which implies that the two maps are inverses of each other. Hence, there is a one-to-one correspondence between matchings of $K_{m,n}$ and deterministic alignments, as desired. $\square$

### B.3    PROOF OF PROPOSITION 2

*Proof of Proposition 2.* Let the feature dissimilarity matrix $C \in \mathbb{R}^{m \times n}$ be constructed with discrepancy $d(\cdot, \cdot) = 1 - \rho(\cdot, \cdot)$, where $\rho$ is the Pearson correlation. In our deterministic model, a feasible $T \in \{0, 1\}^{m \times n} \cap \mathcal{T}(\mathcal{X}_1, \mathcal{X}_2)$ encodes a deterministic matching between $X_1$ and $X_2$. Given an alignment matrix, define the alignment set

$$\lambda(T) \triangleq \{(i, j) \in V_1 \times V_2 : T_{ij} = 1\} \cup \{(i, \emptyset) \in V_1 \times \emptyset : \sum_j T_{ij} = 0\} \cup \{(\emptyset, j) \in \emptyset \times V_2 : \sum_i T_{ij} = 0\}$$

Conversely, any alignment $\lambda$ as defined in Mora et al. (2018a) induces such a feasible $T$.

We prove that when $\alpha = 1$, the cost functional in (1) is equal to (3) up to an additive constant, so that their sets of minimizers are equal.

First, with $\alpha = 1$, the first two terms of Equation (1) collapses to

$$\langle T, A_1 (C \odot T) A_2 \rangle = \sum_{i,j} T_{ij} \sum_{\hat{i} \in \mathcal{N}_{\mathcal{X}_1}(i)} \sum_{\hat{j} \in \mathcal{N}_{\mathcal{X}_2}(j)} C_{\hat{i}\hat{j}} T_{\hat{i}\hat{j}},$$

where $\mathcal{N}_{\mathcal{X}}(i)$ denote the set of 1-hop neighbors of $i$ in $\mathcal{X}$. For each aligned pair of species $x = (i, j) \in \lambda(T)$, the inner double sum ranges over the set of neighbor pairings $\Lambda_x \triangleq \{(\alpha, \beta) \in \lambda(T) : \alpha \in \mathcal{N}_{G_1}(i), \beta \in \mathcal{N}_{G_2}(j)\}$. Thus

$$\langle T, A_1 (C \odot T) A_2 \rangle = \sum_{x \in \lambda(T)} \sum_{(\alpha,\beta) \in \Lambda_x} C_{\alpha\beta} = \sum_{x \in \lambda(T)} \sum_{(\alpha,\beta) \in \Lambda_x} (1 - \rho(\alpha, \beta)), \quad (9)$$

which is exactly the first term in (3).

Next, for some alignment $x = (i, j) \in \lambda(T)$, write the "matched neighbor count" as

$$(A_1 T A_2)_{ij} = |\{(\alpha, \beta) \in \lambda_x\}|.$$

Then, the penalty matrix $\Xi$ in (1) gives, entrywise,

$$\begin{aligned} \Xi_{ij}(T) &= -\epsilon T_{ij} (A_1 T A_2)_{ij} \\ &\propto \epsilon \left( \max \left\{ (A_1 \mathbb{1}_m)_i, (A_2 \mathbb{1}_n)_j \right\} - T_{ij}(A_1 T A_2)_{ij} \right) \\ &= \epsilon \left( \max \left\{ ((A_1 \mathbb{1}_m)_i - T_{ij}(A_1 T A_2)_{ij}, (A_2 \mathbb{1}_n)_j - T_{ij}(A_1 T A_2)_{ij} \right\} \right) \\ &= \epsilon \cdot \max \left\{ k_{x_\alpha}, k_{x_\beta} \right\}, \end{aligned}$$

where $k_{x_\alpha}$ (and $k_{x_\beta}$) is the number of neighbors of $\alpha$ (and $\beta$) that are not paired with a neighbor of $\beta$ (and $\alpha$) (Mora et al., 2018a, Alignment algorithm section of Supplementary Methods). Therefore,

$$\Xi(T) \propto \sum_{x \in \lambda(T)} (1 - \epsilon) \max(k_x^\alpha, k_x^\beta) = \sum_{x \in \lambda(T)} \xi_x. \quad (10)$$

Combining Equations (9) and (10), we have

$$\begin{aligned} & g(T, C, 1, \epsilon) \\ &= \langle T, A_1 (C \odot T) A_2 \rangle + \Xi(T) \\ &\propto \sum_{x \in \lambda(T)} \sum_{(\alpha,\beta) \in \Lambda_x} 1 - \rho(\alpha, \beta) + \sum_{x \in \lambda(T)} \xi_x \\ &= \sum_{x \in \lambda(T)} \left( \sum_{(\alpha,\beta) \in \Lambda_x} (1 - \rho(\alpha, \beta)) + \xi_x \right), \end{aligned}$$

as desired. $\qquad \square$

## C  BOUNDED ACCUMULATIVE ASYMMETRICAL ERROR (AAE): ASSUMPTION 1

For completeness, we briefly recall the setup we are working on. We work with matrices $T \in \mathbb{R}^{m \times n}$ with strictly positive entries and the entropic generator

$$h(X) \; = \; \sum_{i,j} x_{ij} \log x_{ij}, \qquad X = (x_{ij})_{i,j}, \; x_{ij} > 0.$$

The associated Bregman divergence is

$$D_h(X \| Y) \; = \; \sum_{i,j} \left( x_{ij} \log \frac{x_{ij}}{y_{ij}} - x_{ij} + y_{ij} \right),$$

defined for strictly positive matrices $X, Y$.

We consider the row and column constraint sets

$$C_1(\mu) \; := \; \{T \in \mathbb{R}_+^{m \times n} : T 1_n \preceq \mu\}, \qquad C_2(\nu) \; := \; \{T \in \mathbb{R}_+^{m \times n} : T^\top 1_m \preceq \nu\},$$

for fixed vectors $\mu \in \mathbb{R}_+^m$, $\nu \in \mathbb{R}_+^n$, where $1_k$ is the all-ones vector and $\preceq$ denotes component-wise inequality.

We are also going to use the following facts and assumptions:

- $\mu_i > 0$ and $\nu_j > 0$ for all $i, j$, and $\sum_i \mu_i < \infty$, $\sum_j \nu_j < \infty$. We have worked with positive margins only. This is a working assumption that is maintained throughout the paper.

- $C_\varepsilon \in \mathbb{R}^{m \times n}$ and $A_1 \in \mathbb{R}^{m \times m}$, $A_2 \in \mathbb{R}^{n \times n}$ are fixed finite matrices, with $C_\varepsilon := C - \varepsilon 1_m 1_n^\top$, for some given $C$ and $\varepsilon \geq 0$.

- The initial iterate $T^{(0)}$ satisfies $T^{(0)} > 0$ and $T^{(0)} \in C_2(\nu)$. This condition holds automatically with our initialization scheme $T^{(0)} = \frac{1}{mn} 1_m 1_n^\top$

We slightly rewrite our proposed algorithm in (2) to enhance exposition and clarity of proof. Indexed by $k \geq 1$, (2) can be rewriten as:

$$\begin{aligned}
U_{\text{row}}^{(k)} &:= T^{(k-1)} \odot \exp(-\gamma_k Q^{(k-1)}), \\
\hat{T}^{(k)} &:= P_{C_1(\mu)}\big(U_{\text{row}}^{(k)}\big), \\
U_{\text{col}}^{(k)} &:= \hat{T}^{(k)} \odot \exp(-\gamma_k Q'^{(k)}), \\
T^{(k+1)} &:= P_{C_2(\nu)}\big(U_{\text{col}}^{(k)}\big),
\end{aligned} \qquad (11)$$

where $P_{C_1(\mu)}$ and $P_{C_2(\nu)}$ denote the KL projections onto $C_1(\mu)$ and $C_2(\nu)$, respectively, and

$$\begin{aligned}
Q^{(k)} &:= \alpha A_1 \big(C_\varepsilon \odot T^{(k-1)}\big) A_2 + \tfrac{1}{2}(1 - \alpha) C_\varepsilon, \\
Q'^{(k)} &:= \alpha C_\varepsilon \odot \big(A_1 \hat{T}^{(k)} A_2\big) + \tfrac{1}{2}(1 - \alpha) C_\varepsilon,
\end{aligned} \qquad (12)$$

for some fixed $\alpha \in [0, 1]$. The step sizes $\gamma_k > 0$ will be specified below.

The multiplicative updates in (11) involve exponentials of finite matrices and start from $T^{(0)} > 0$. Thus, they maintain strict positivity at each iteration. Similarly, KL projections onto $C_1(\mu)$ and $C_2(\nu)$ with positive inputs yield strictly positive solutions: the scalar function

$$t \; \mapsto \; t \log(t/u) - t + u$$

has derivative $\log(t/u)$, which tends to $-\infty$ as $t \to 0^+$; since the constraints are of the form "$\leq$" with positive right-hand side, the minimizer cannot be at zero. Thus all iterates $T^{(k)}, \hat{T}^{(k)}, U_{\text{row}}^{(k)}, U_{\text{col}}^{(k)}$ have strictly positive entries at any given iteration $k$, yet no uniform lower bound is available.

For the adjacency matrices and cost function, we define the row/column norms:

$$B_1 := \max_{1 \leq i \leq m} \sum_{p=1}^{m} |(A_1)_{ip}|, \qquad B_2 := \max_{1 \leq j \leq n} \sum_{q=1}^{n} |(A_2)_{qj}|, \qquad \|C_\varepsilon\|_\infty := \max_{i,j} |(C_\varepsilon)_{ij}|.$$

Finally, we introduce the global bound

$$T_{\max} := \max\{\max_i \mu_i, \max_j \nu_j\} < \infty.$$

**Lemma 2** (Uniform bound on $Q^{(k)}$ and $Q'^{(k)}$). *Let $Q^{(k)}$ and $Q'^{(k)}$ defined in* (12)*, and let*

$$L_Q := \|C_\varepsilon\|_\infty T_{\max} \left( \alpha B_1 B_2 + \frac{1-\alpha}{2} \right).$$

*Thus, for all iterations $k \geq 0$ and all indices $i, j$,*

$$|Q_{ij}^{(k)}| \leq L_Q, \qquad |Q_{ij}'^{(k)}| \leq L_Q.$$

*Proof.* We first bound $Q^{(k)}$. Define

$$M^{(k)} := C_\varepsilon \odot T^{(k-1)}, \qquad M_{pq}^{(k)} = (C_\varepsilon)_{pq} T_{pq}^{(k-1)}.$$

Since $T^{(k-1)} \in C_2(\nu)$, for each column $j$ we have $\sum_i T_{ij}^{(k-1)} \leq \nu_j$, hence $0 < T_{ij}^{(k-1)} \leq \nu_j \leq T_{\max}$. Thus

$$|M_{pq}^{(k)}| \leq \|C_\varepsilon\|_\infty T_{\max} \quad \text{for all } p, q.$$

We then compute

$$(A_1 M^{(k)} A_2)_{ij} = \sum_{p,q} (A_1)_{ip} M_{pq}^{(k)} (A_2)_{qj}.$$

Using the bound on $M^{(k)}$, we obtain

$$\left| (A_1 M^{(k)} A_2)_{ij} \right| \leq \sum_{p,q} |(A_1)_{ip}| \, |M_{pq}^{(k)}| \, |(A_2)_{qj}|$$

$$\leq \|C_\varepsilon\|_\infty T_{\max} \sum_{p,q} |(A_1)_{ip}| \, |(A_2)_{qj}|$$

$$= \|C_\varepsilon\|_\infty T_{\max} \left( \sum_p |(A_1)_{ip}| \right) \left( \sum_q |(A_2)_{qj}| \right)$$

$$\leq \|C_\varepsilon\|_\infty T_{\max} B_1 B_2.$$

The second term in (12) satisfies

$$\left| \tfrac{1}{2}(1 - \alpha)(C_\varepsilon)_{ij} \right| \leq \tfrac{1}{2}(1 - \alpha)\|C_\varepsilon\|_\infty.$$

Therefore,

$$|Q_{ij}^{(k)}| = \left| \alpha (A_1 M^{(k)} A_2)_{ij} + \tfrac{1}{2}(1 - \alpha)(C_\varepsilon)_{ij} \right|$$

$$\leq \alpha \|C_\varepsilon\|_\infty T_{\max} B_1 B_2 + \tfrac{1}{2}(1 - \alpha)\|C_\varepsilon\|_\infty.$$

By the definition of $L_Q$, we conclude that $|Q_{ij}^{(k)}| \leq L_Q$ for all $i, j, k$.

We now bound $Q'^{(k)}$. Define

$$N^{(k)} := A_1 \hat{T}^{(k)} A_2.$$

Since $\hat{T}^{(k)} \in C_1(\mu)$, for each row $i$ we have $\sum_j \hat{T}_{ij}^{(k)} \leq \mu_i$, hence $0 < \hat{T}_{ij}^{(k)} \leq \mu_i \leq T_{\max}$, and thus

$$|N_{ij}^{(k)}| = \left| \sum_{p,q} (A_1)_{ip} \hat{T}_{pq}^{(k)} (A_2)_{qj} \right|$$

$$\leq \sum_{p,q} |(A_1)_{ip}| \, \hat{T}_{pq}^{(k)} \, |(A_2)_{qj}|$$

$$\leq T_{\max} \sum_{p,q} |(A_1)_{ip}| \, |(A_2)_{qj}|$$

$$\leq T_{\max} B_1 B_2.$$

Hence

$$|(C_\varepsilon \odot N^{(k)})_{ij}| = |(C_\varepsilon)_{ij}| \, |N^{(k)}_{ij}| \le \|C_\varepsilon\|_\infty T_{\max} B_1 B_2,$$

and therefore

$$|Q'^{(k)}_{ij}| \le \alpha \|C_\varepsilon\|_\infty T_{\max} B_1 B_2 + \tfrac{1}{2}(1-\alpha)\|C_\varepsilon\|_\infty \le L_Q.$$

This establishes the claimed bounds for both $Q^{(k)}$ and $Q'^{(k)}$. $\qquad\square$

**Lemma 3** (Local cubic asymmetry of the scalar entropy). *For scalars $u, v > 0$, define*

$$D(u\|v) := u \log \frac{u}{v} - u + v, \qquad A(u,v) := D(u\|v) - D(v\|u).$$

*Let $\delta_0 = \tfrac{1}{2}$ and $C_0 = 2$. Then, for any $u, v > 0$ satisfying*

$$|u - v| \le \delta_0 \min(u, v),$$

*it holds that*

$$|A(u,v)| \le C_0 \frac{|u-v|^3}{\min(u,v)^2}.$$

*Proof.* We first compute $A(u,v)$ in closed form. From the definition,

$$D(u\|v) = u \log \frac{u}{v} - u + v,$$
$$D(v\|u) = v \log \frac{v}{u} - v + u,$$

so

$$\begin{aligned}
A(u,v) &= D(u\|v) - D(v\|u) \\
&= u \log \frac{u}{v} - u + v - \left(v \log \frac{v}{u} - v + u\right) \\
&= u \log \frac{u}{v} + v \log \frac{u}{v} - 2(u - v) \\
&= (u + v) \log \frac{u}{v} - 2(u - v).
\end{aligned}$$

Without loss of generality, assume $u \ge v$. We may then write $u = rv$ with $r \ge 1$. In that case,

$$u - v = (r-1)v, \qquad \frac{u}{v} = r,$$

and

$$A(u,v) = v f(r), \qquad f(r) := (r+1)\log r - 2(r-1).$$

We now study $f(r)$ near $r = 1$. Direct differentiation yields

$$f(1) = 0, \qquad f'(r) = \log r + \frac{1}{r} - 1, \quad f'(1) = 0,$$

$$f''(r) = \frac{1}{r} - \frac{1}{r^2} = \frac{r-1}{r^2}, \quad f''(1) = 0,$$

and

$$f^{(3)}(r) = \frac{d}{dr}\left(\frac{r-1}{r^2}\right) = \frac{2-r}{r^3}.$$

The condition $|u - v| \le \delta_0 \min(u, v)$ with $\delta_0 = 1/2$ and $u \ge v$ becomes

$$|u - v| = (r-1)v \le \frac{1}{2}v \quad \Rightarrow \quad |r - 1| \le \frac{1}{2},$$

that is, $r \in [1/2, 3/2]$. For such $r$,

$$|f^{(3)}(r)| = \frac{|2-r|}{r^3} \le \frac{2 - 1/2}{(1/2)^3} = \frac{3/2}{1/8} = 12.$$

Let $M_3 := 12$.

By Taylor's theorem with Lagrange remainder around $r = 1$, there exists a point $\xi_r$ between 1 and $r$ such that

$$f(r) = f(1) + f'(1)(r - 1) + \frac{1}{2}f''(1)(r - 1)^2 + \frac{1}{6}f^{(3)}(\xi_r)(r - 1)^3.$$

Since $f(1) = f'(1) = f''(1) = 0$, this simplifies to

$$f(r) = \frac{1}{6}f^{(3)}(\xi_r)(r - 1)^3.$$

Therefore, for all $r$ with $|r - 1| \leq 1/2$,

$$|f(r)| \leq \frac{M_3}{6}|r - 1|^3 = 2|r - 1|^3.$$

Recalling that $r = u/v$ and $\min(u, v) = v$ in the case $u \geq v$, we note that

$$|r - 1| = \left|\frac{u}{v} - 1\right| = \frac{|u - v|}{v} = \frac{|u - v|}{\min(u, v)}.$$

It follows that

$$\begin{aligned}|A(u, v)| &= v|f(r)| \\ &\leq 2v|r - 1|^3 \\ &= 2v\left(\frac{|u - v|}{v}\right)^3 \\ &= 2\frac{|u - v|^3}{v^2} \\ &= 2\frac{|u - v|^3}{\min(u, v)^2}.\end{aligned}$$

This proves the desired inequality when $u \geq v$.

If $u < v$, then the condition $|u - v| \leq \delta_0 \min(u, v)$ is symmetric in $(u, v)$, and $A(u, v) = -A(v, u)$. Hence the same bound holds in this case as well. The lemma is therefore proved with $\delta_0 = \frac{1}{2}$ and $C_0 = 2$. $\qquad\square$

**Remark C.1.** *For matrices $X = (x_{ij})$ and $Y = (y_{ij})$ with strictly positive entries, if*

$$|x_{ij} - y_{ij}| \leq \delta_0 \min(x_{ij}, y_{ij}) \quad \forall i, j,$$

*then, with $D$ and $A$ as in Lemma 3,*

$$D_h(X\|Y) - D_h(Y\|X) = \sum_{i,j} A(x_{ij}, y_{ij}),$$

*and Lemma 3 gives*

$$\left|D_h(X\|Y) - D_h(Y\|X)\right| \leq C_0 \sum_{i,j} \frac{|x_{ij} - y_{ij}|^3}{\min(x_{ij}, y_{ij})^2}.$$

**Lemma 4** (Marginal stability of the $C_2(\nu)$ block)**.** *Let $L_Q$ be as in Lemma 2, and suppose the iterates $(T^{(k)}, \hat{T}^{(k)})$ are generated by the scheme (11)–(12) with strictly positive initial point $T^{(0)} \in C_2(\nu)$. Assume the step sizes satisfy*

$$0 < \gamma_k \leq \frac{1}{4L_Q} \quad \text{for all } k.$$

*Then there exists a constant*

$$C_g := 40L_Q$$

*such that, for all $k \geq 0$ and all indices $i, j$,*

$$\left|T_{ij}^{(k+1)} - \hat{T}_{ij}^{(k)}\right| \leq C_g\,\gamma_k\,\min\left(T_{ij}^{(k+1)}, \hat{T}_{ij}^{(k)}\right).$$

*Proof.* Fix an iteration index $k \geq 1$ and, to simplify notation, denote

$$T^- := T^{(k-1)}, \quad U_{\text{row}} := U_{\text{row}}^{(k)}, \quad \hat{T} := \hat{T}^{(k)}, \quad U_{\text{col}} := U_{\text{col}}^{(k)}, \quad T^+ := T^{(k+1)},$$

and

$$Q := Q^{(k-1)}, \quad Q' := Q'^{(k)}, \quad \gamma := \gamma_k.$$

By construction, $T^-, T^+ \in C_2(\nu)$ and all matrices have strictly positive entries. Lemma 2 gives $|Q_{ij}| \leq L_Q$ and $|Q'_{ij}| \leq L_Q$.

We first control the column sums after the row update. The multiplicative step is given by

$$U_{\text{row},ij} = T^-_{ij} e^{-\gamma Q_{ij}}.$$

For each column $j$,

$$S_j(U_{\text{row}}) := \sum_i U_{\text{row},ij} = \sum_i T^-_{ij} e^{-\gamma Q_{ij}}$$

$$\leq e^{\gamma L_Q} \sum_i T^-_{ij} \leq e^{\gamma L_Q} \nu_j,$$

where we used $|Q_{ij}| \leq L_Q$ and $T^- \in C_2(\nu)$.

Next, $\hat{T}$ is the KL projection of $U_{\text{row}}$ onto $C_1(\mu)$, which is row-wise. For each row $i$, the projection problem is

$$\min_{t_i \geq 0} \sum_j \left( t_{ij} \log \frac{t_{ij}}{u_{ij}} - t_{ij} + u_{ij} \right) \quad \text{s.t.} \quad \sum_j t_{ij} \leq \mu_i,$$

where $u_{ij} := U_{\text{row},ij}$. The KKT conditions show that

$$\hat{T}_{ij} = \alpha_i U_{\text{row},ij}, \qquad \alpha_i := \min\left(1, \frac{\mu_i}{\sum_\ell U_{\text{row},i\ell}}\right) \in (0, 1].$$

In particular, $\hat{T}_{ij} \leq U_{\text{row},ij}$ for all $i, j$. Consequently, for each column $j$,

$$S_j(\hat{T}) := \sum_i \hat{T}_{ij} \leq \sum_i U_{\text{row},ij} = S_j(U_{\text{row}}) \leq e^{\gamma L_Q} \nu_j.$$

We now consider the second multiplicative step

$$U_{\text{col},ij} = \hat{T}_{ij} e^{-\gamma Q'_{ij}}.$$

For each column $j$,

$$S_j(U_{\text{col}}) := \sum_i U_{\text{col},ij} = \sum_i \hat{T}_{ij} e^{-\gamma Q'_{ij}}$$

$$\leq e^{\gamma L_Q} \sum_i \hat{T}_{ij}$$

$$\leq e^{2\gamma L_Q} \nu_j,$$

using $|Q'_{ij}| \leq L_Q$ and the bound on $S_j(\hat{T})$.

For $x \in [0, 1]$, the function $g(x) := 1 + 2x - e^x$ satisfies $g(0) = 0$ and $g'(x) = 2 - e^x$, hence $g(x) \geq 0$ on $[0, 1]$. Thus $e^x \leq 1 + 2x$ whenever $x \in [0, 1]$. Since $\gamma L_Q \leq 1/(4L_Q) \cdot L_Q = 1/4$, we have $2\gamma L_Q \leq 1/2 < 1$ and therefore

$$e^{2\gamma L_Q} \leq 1 + 4\gamma L_Q.$$

It follows that

$$S_j(U_{\text{col}}) \leq (1 + 4\gamma L_Q)\nu_j \quad \text{for all } j.$$

We now examine the KL projection of $U_{\text{col}}$ onto $C_2(\nu)$. For a fixed column $j$, we solve

$$\min_{t_{\cdot j} \geq 0} \sum_i \left( t_{ij} \log \frac{t_{ij}}{u_{ij}} - t_{ij} + u_{ij} \right) \quad \text{s.t.} \quad \sum_i t_{ij} \leq \nu_j,$$

where $u_{ij} := U_{\text{col},ij}$. The Lagrangian is

$$\mathcal{L}(t, \lambda) = \sum_i \left( t_{ij} \log \frac{t_{ij}}{u_{ij}} - t_{ij} + u_{ij} \right) + \lambda \left( \sum_i t_{ij} - \nu_j \right), \quad \lambda \geq 0.$$

First-order optimality for $t_{ij} > 0$ yields

$$\frac{\partial \mathcal{L}}{\partial t_{ij}} = \log \frac{t_{ij}}{u_{ij}} + \lambda = 0 \quad \Rightarrow \quad t_{ij} = u_{ij} e^{-\lambda}.$$

Thus the minimizer has the form

$$T_{ij}^+ = \beta_j U_{\text{col},ij}, \quad \text{with} \quad \beta_j := e^{-\lambda} > 0.$$

The complementary slackness and feasibility conditions imply

$$\sum_i T_{ij}^+ \leq \nu_j, \quad \lambda \geq 0, \quad \lambda \left( \sum_i T_{ij}^+ - \nu_j \right) = 0.$$

Hence either the constraint is inactive and $\beta_j = 1$, or the constraint is active and

$$\sum_i T_{ij}^+ = \beta_j \sum_i U_{\text{col},ij} = \nu_j \quad \Rightarrow \quad \beta_j = \frac{\nu_j}{S_j(U_{\text{col}})}.$$

In all cases,

$$T_{ij}^+ = \beta_j U_{\text{col},ij}, \quad \beta_j = \min \left( 1, \frac{\nu_j}{S_j(U_{\text{col}})} \right).$$

If $S_j(U_{\text{col}}) \leq \nu_j$, then $\beta_j = 1$ and $T_{ij}^+ = U_{\text{col},ij}$. If $S_j(U_{\text{col}}) > \nu_j$, then the bound $S_j(U_{\text{col}}) \leq (1 + 4\gamma L_Q)\nu_j$ implies

$$\beta_j = \frac{\nu_j}{S_j(U_{\text{col}})} \geq \frac{\nu_j}{(1 + 4\gamma L_Q)\nu_j} = \frac{1}{1 + 4\gamma L_Q}.$$

In that case,

$$1 - \beta_j = 1 - \frac{\nu_j}{S_j(U_{\text{col}})} = \frac{S_j(U_{\text{col}}) - \nu_j}{S_j(U_{\text{col}})} \leq \frac{(1 + 4\gamma L_Q)\nu_j - \nu_j}{\nu_j} = 4\gamma L_Q.$$

Consequently, in both cases the bound

$$|1 - \beta_j| \leq 4\gamma L_Q$$

holds. Moreover, if $\gamma \leq 1/(4L_Q)$, then $4\gamma L_Q \leq 1$ and

$$\beta_j \geq \frac{1}{1 + 4\gamma L_Q} \geq \frac{1}{2}.$$

Using $T_{ij}^+ = \beta_j U_{\text{col},ij}$, we deduce

$$|T_{ij}^+ - U_{\text{col},ij}| = U_{\text{col},ij} |1 - \beta_j| \leq 4L_Q \gamma\, U_{\text{col},ij}.$$

On the other hand, $\beta_j \in [1/2, 1]$ implies $T_{ij}^+ \leq U_{\text{col},ij}$ and $T_{ij}^+ \geq \frac{1}{2} U_{\text{col},ij}$, whence

$$\min(T_{ij}^+, U_{\text{col},ij}) = T_{ij}^+ \geq \frac{1}{2} U_{\text{col},ij},$$

and the previous inequality may also be written as

$$|T_{ij}^+ - U_{\text{col},ij}| \leq 8L_Q \gamma\, \min(T_{ij}^+, U_{\text{col},ij}).$$

We now relate $U_{\text{col}}$ and $\hat{T}$. From the definition,

$$U_{\text{col},ij} = \hat{T}_{ij} \exp(-\gamma Q'_{ij}),$$

and by Lemma 2, $|Q'_{ij}| \leq L_Q$. Thus $|\gamma Q'_{ij}| \leq \gamma L_Q \leq 1/4$. For any $x$ with $|x| \leq 1/2$, the function $e^{-x}$ satisfies $|e^{-x} - 1| \leq e^{|x|}|x| \leq e^{1/2}|x| < 2|x|$, hence

$$|e^{-\gamma Q'_{ij}} - 1| \leq 2|\gamma Q'_{ij}| \leq 2\gamma L_Q.$$

This yields

$$|U_{\text{col},ij} - \hat{T}_{ij}| = \hat{T}_{ij}|e^{-\gamma Q'_{ij}} - 1| \leq 2L_Q\gamma\,\hat{T}_{ij}.$$

Furthermore, since $|\gamma Q'_{ij}| \leq 1/4$, we have $e^{-1/4} \leq e^{-\gamma Q'_{ij}} \leq e^{1/4}$, and the numerical estimates $e^{1/4} < 2$ and $e^{-1/4} > 1/2$ imply

$$\frac{1}{2}\hat{T}_{ij} \leq U_{\text{col},ij} \leq 2\hat{T}_{ij}.$$

Introduce the quantity

$$m_{ij} := \min(T^+_{ij}, \hat{T}_{ij}).$$

From the inequality $U_{\text{col},ij} \geq \frac{1}{2}\hat{T}_{ij}$ and the bound $T^+_{ij} \geq \frac{1}{2}U_{\text{col},ij}$, we obtain

$$T^+_{ij} \geq \frac{1}{2}U_{\text{col},ij} \geq \frac{1}{2} \cdot \frac{1}{2}\hat{T}_{ij} = \frac{1}{4}\hat{T}_{ij}.$$

Thus $\hat{T}_{ij} \leq 4T^+_{ij}$. Combining this with $U_{\text{col},ij} \leq 2\hat{T}_{ij}$ yields $U_{\text{col},ij} \leq 8T^+_{ij}$.

If $m_{ij} = \hat{T}_{ij} \leq T^+_{ij}$, then $\hat{T}_{ij} = m_{ij}$ and $U_{\text{col},ij} \leq 2\hat{T}_{ij} = 2m_{ij}$. If $m_{ij} = T^+_{ij} < \hat{T}_{ij}$, then $\hat{T}_{ij} \leq 4T^+_{ij} = 4m_{ij}$ and $U_{\text{col},ij} \leq 8T^+_{ij} = 8m_{ij}$. In all cases,

$$\hat{T}_{ij} \leq 4m_{ij}, \qquad U_{\text{col},ij} \leq 8m_{ij}.$$

We now combine the previous inequalities:

$$\begin{aligned}
|T^+_{ij} - \hat{T}_{ij}| &\leq |T^+_{ij} - U_{\text{col},ij}| + |U_{\text{col},ij} - \hat{T}_{ij}| \\
&\leq 4L_Q\gamma\,U_{\text{col},ij} + 2L_Q\gamma\,\hat{T}_{ij} \\
&\leq 4L_Q\gamma \cdot 8m_{ij} + 2L_Q\gamma \cdot 4m_{ij} \\
&= (32 + 8)L_Q\gamma\,m_{ij} \\
&= 40L_Q\gamma\,\min(T^+_{ij}, \hat{T}_{ij}).
\end{aligned}$$

Defining $C_g := 40L_Q$ gives the desired inequality

$$|T^+_{ij} - \hat{T}_{ij}| \leq C_g\gamma\,\min(T^+_{ij}, \hat{T}_{ij}).$$

Since the argument holds for every $k \geq 1$ and all $i, j$, the lemma is proved. $\qquad\square$

**Theorem 2** (Summability of the asymmetric error for entropic $h$). *Let $h(X) = \sum_{i,j} x_{ij} \log x_{ij}$ and $D_h$ be its associated Bregman divergence. Let $(T^{(k)}, \hat{T}^{(k)})$ be the sequence generated by the scheme (11)–(12) under the setup and assumptions stated above. Define, for each $k \geq 0$,*

$$\Delta_k := D_h\big(\hat{T}^{(k)}, T^{(k+1)}\big) - D_h\big(T^{(k+1)}, \hat{T}^{(k)}\big).$$

*Assume the step sizes satisfy*

$$0 < \gamma_k \leq \bar{\gamma} := \frac{1}{80L_Q} \quad \text{for all } k, \qquad \sum_{k=0}^{\infty}\gamma_k^3 < \infty.$$

*Then*

$$\sum_{k=0}^{\infty}|\Delta_k| < \infty.$$

*In particular, the accumulative asymmetric error*

$$\sum_{k=0}^{\infty}\big(D_h(\hat{T}^{(k)}, T^{(k+1)}) - D_h(T^{(k+1)}, \hat{T}^{(k)})\big)$$

*is absolutely convergent, i.e., Assumption 1 holds for the entropic generator $h$ under these conditions.*

*Proof.* Since $\hat{T}^{(k)} \in C_1(\mu)$ for all $k$, we have

$$\sum_{i,j} \hat{T}_{ij}^{(k)} = \sum_i (\hat{T}^{(k)} 1_n)_i \leq \sum_i \mu_i =: M < \infty.$$

By Lemma 4 with $C_g = 40L_Q$, for each $k, i, j$,

$$\left| T_{ij}^{(k+1)} - \hat{T}_{ij}^{(k)} \right| \leq C_g \gamma_k \, \min\!\left(T_{ij}^{(k+1)}, \hat{T}_{ij}^{(k)}\right).$$

Since $\gamma_k \leq \bar{\gamma} = 1/(80L_Q)$, we obtain

$$C_g \gamma_k = 40L_Q \gamma_k \leq 40L_Q \cdot \frac{1}{80L_Q} = \frac{1}{2}.$$

Let $\delta_0 = \frac{1}{2}$ be the constant from Lemma 3. Then

$$\left| T_{ij}^{(k+1)} - \hat{T}_{ij}^{(k)} \right| \leq \delta_0 \min\!\left(T_{ij}^{(k+1)}, \hat{T}_{ij}^{(k)}\right) \quad \text{for all } i, j, k.$$

Define

$$u_{ij}^{(k)} := \hat{T}_{ij}^{(k)}, \qquad v_{ij}^{(k)} := T_{ij}^{(k+1)}, \qquad m_{ij}^{(k)} := \min(u_{ij}^{(k)}, v_{ij}^{(k)}).$$

The inequality above shows that the pair $(u_{ij}^{(k)}, v_{ij}^{(k)})$ satisfies the locality condition of Lemma 3 for all $i, j, k$. Hence Lemma 3 gives

$$\left| D(u_{ij}^{(k)} \| v_{ij}^{(k)}) - D(v_{ij}^{(k)} \| u_{ij}^{(k)}) \right| \leq C_0 \frac{|u_{ij}^{(k)} - v_{ij}^{(k)}|^3}{(m_{ij}^{(k)})^2},$$

where $C_0 = 2$. Using Lemma 4 again,

$$|u_{ij}^{(k)} - v_{ij}^{(k)}| = |\hat{T}_{ij}^{(k)} - T_{ij}^{(k+1)}| \leq C_g \gamma_k \, m_{ij}^{(k)}.$$

It follows that

$$\frac{|u_{ij}^{(k)} - v_{ij}^{(k)}|^3}{(m_{ij}^{(k)})^2} \leq C_g^3 \gamma_k^3 m_{ij}^{(k)}.$$

Therefore,

$$\left| D(u_{ij}^{(k)} \| v_{ij}^{(k)}) - D(v_{ij}^{(k)} \| u_{ij}^{(k)}) \right| \leq C_0 C_g^3 \gamma_k^3 m_{ij}^{(k)}.$$

The matrix-level asymmetric error can be written as

$$\Delta_k = D_h(\hat{T}^{(k)}, T^{(k+1)}) - D_h(T^{(k+1)}, \hat{T}^{(k)}) = \sum_{i,j} \left( D(u_{ij}^{(k)} \| v_{ij}^{(k)}) - D(v_{ij}^{(k)} \| u_{ij}^{(k)}) \right).$$

Taking absolute values and applying the previous bound yields

$$|\Delta_k| \leq \sum_{i,j} \left| D(u_{ij}^{(k)} \| v_{ij}^{(k)}) - D(v_{ij}^{(k)} \| u_{ij}^{(k)}) \right| \leq C_0 C_g^3 \gamma_k^3 \sum_{i,j} m_{ij}^{(k)}.$$

Since $m_{ij}^{(k)} \leq u_{ij}^{(k)} = \hat{T}_{ij}^{(k)}$, we have

$$\sum_{i,j} m_{ij}^{(k)} \leq \sum_{i,j} \hat{T}_{ij}^{(k)} \leq M,$$

where $M := \sum_i \mu_i < \infty$. Thus

$$|\Delta_k| \leq C_* \gamma_k^3, \qquad C_* := C_0 C_g^3 M.$$

Finally, by the assumption $\sum_{k=0}^{\infty} \gamma_k^3 < \infty$, we obtain

$$\sum_{k=0}^{\infty} |\Delta_k| \leq C_* \sum_{k=0}^{\infty} \gamma_k^3 < \infty.$$

This establishes the claimed absolute convergence of the series $\sum_k |\Delta_k|$, and hence Assumption 1 holds for the entropic generator $h$ under the stated conditions. $\qquad\square$

