# OpenReview forum: "Wasserstein Motifs: Non-deterministic Alignment of Ecological Networks"
_ICLR.cc/2026/Conference — Submitted to ICLR 2026_

### Official Review · Reviewer_j2Bj · 2025-10-31

**Soundness:** 3
**Presentation:** 3
**Contribution:** 3
**Rating:** 6
**Confidence:** 4

**Summary:**

The paper formalizes food-web, an ecological network, alignment via a motif-based, non-deterministic OT objective. The core optimization (Eq. 1) fuses a zeroth-order feature term with a first-order topology term and introduces a “self-alignment” penalty to avoid trivial mappings. The paper connects an influential ecological alignment heuristic to a special case of their objective (α=1), propose a KL-BAPG variant with Dykstra projections to compute transport plans under ≤ marginal constraints, and leverage the learned transport to define “non-deterministic backbones” and a fractional triangle-style transitivity score across networks. Experiments on 129 Sub-Saharan mammal food webs report faster runtimes than simulated annealing and qualitatively more ecologically coherent alignments than FGW/partial-FGW baselines, plus meta-alignment analyses by diet/biomass groups and top-k backbone diagnostics).

**Strengths:**

1. The paper gives clear formalization of a long-standing ecological heuristic. Proposition 2 shows that in the deterministic case with $\alpha$=1 and Pearson-based feature discrepancy, the objective (Eq. 1) matches the annealing score of Mora et al. up to an additive constant, hence they share minimizers.  This provides a principled geometric footing for that heuristic within their OT-style framework.

2. The formulation presented in the paper naturally supports many-to-many correspondences via a transport plan $T$ under $\leq$ marginal constraints. The objective splits into a zeroth-order feature term <T, C> by using motif-role features and a first-order, adjacency-aware term <T,A_1 (T⊙C) A_2> that promotes neighborhood-consistent matches. A self-alignment penalty controlled by ε discourages degenerate self-mass solutions, improving ecological plausibility. Algorithm 1’s KL-BAPG with Dykstra projections gives a lightweight routine and enforces the ≤ marginal constraints by simple multiplicative rescaling, making the method easy to implement and numerically stable.


3. The paper generalizes the backbone concept to non-deterministic mappings by introducing a role-similarity score and a fractional, triangle-based transitivity. Empirically, the resulting top-k backbones are more connected and more transitive than a than a null model at small k shown in Fig. 12.

**Weaknesses:**

1.	The objective’s first-order term and the linear feature term look close to FGW-style formulations. Without fair, feature-matched baselines, for example, FGW using the same motif-role features, FGW with motifs+traits, it’s hard to isolate what is new: the objective design, the penalty, or simply the features.
2.	The paper does not systematically study the behavior or provide robust defaults for the tradeoff parameter $\alpha$, self-alignment regularization parameter $\varepsilon$  or  step-size $\gamma$ , nor how these interact.
3.	The evaluation is potentially unfair and weak. The paper compares its method using motif-role features against FGW/Partial-FGW baselines using trait features, so improvements may reflect feature choice, not the objective/solver.

**Questions:**

1. How do results change when FGW uses the same motif features and traits+motifs? Also, could the authors do ablation study of the following: (1) first-order term off, (2) self-alignment penalty off?

2.	If you keep the graph directed in the first-order term (e.g., separate in/out adjacency), do alignments/backbones improve?

3. see weakness 2.

---

> ### Author Response · Authors · 2025-11-22
> **Response to Reviewer j2Bj (1)**
>
> > The objective’s first-order term and the linear feature term look close to FGW-style formulations. Without fair, feature-matched baselines, for example, FGW using the same motif-role features, FGW with motifs+traits, it’s hard to isolate what is new: the objective design, the penalty, or simply the features.
>
> and
>
> > The evaluation is potentially unfair and weak. The paper compares its method using motif-role features against FGW/Partial-FGW baselines using trait features, so any improvements may reflect the choice of features, rather than the objective/solver.
>
> and
>
> > How do results change when FGW uses the same motif features and traits+motifs?
>
> We thank the reviewer for this insightful comment. Although our objective contains a linear "zeroth-order'' term and a quadratic "first-order'' term, *our formulation is fundamentally different from the Fused Gromov–Wasserstein (FGW) objective* in both (i) modelling assumptions and (ii) objective structure.
>
> **Structural difference.** Our first-order term depends on *edges*, whereas FGW depends on *metrics*. Expanding our first-order term,
> $$
> \langle T, A\_1 (T \odot C) A\_2\rangle
> = \sum\_{i,j,i',j'} (A\_1)\_{ii'} (A\_2)\_{jj'} C\_{i'j'} T\_{ij} T\_{i'j'}.
> $$
> A term contributes *only when both edges exist*:
> $$
> (A\_1)\_{ii'} = 1,\qquad (A\_2)\_{jj'} = 1.
> $$
> Thus, our model rewards alignments whose neighbors are also aligned *and* have low motif-profile dissimilarity. Feature information appears *inside* the structural interaction, producing a localized, feature-aware quadratic term.
>
> In contrast, FGW assumes two metric spaces $(X_1,d_1)$ and $(X_2,d_2)$, with quadratic term
> $$
> \sum_{i,j,i',j'} L \left(d_1(x_i,x_{i'}), d_2(y_j,y_{j'})\right) T_{ij}T_{i'j'},
> $$
> where $L: \mathbb{R}\_+ \times \mathbb{R}\_+ \rightarrow \mathbb{R}$ is a loss function. Here, *every* quadruple contributes regardless of edges; structural information enters only via pairwise distances, and feature similarity never enters the quadratic term. **No choice of $d_1,d_2$ or loss $L$ in FGW can reproduce our edge-filtered, feature-modulated interaction.**
>
> **Feature-matched FGW baselines.**
> To address the reviewer’s concern regarding feature choice, we implemented FGW variants that *explicitly use the same motif-role profiles* as our method:
> - **FGW (traits as features, motifs as metric):** Gower trait distance as features; motif-role dissimilarity as metric.
> - **FGW (motifs as features and metric):** motif-role dissimilarity used for both features and metric.
>
> We also evaluated the vanilla FGW baseline [11]. All baselines were assessed on:
> - L1 alignment discrepancy (Fig. 8),
> - meta-alignment consistency (Fig. 9),
> - backbone quality (Fig. 12).
>
> All FGW variants separate identical vs. non-identical species in the discrepancy test. For meta-alignment, FGW with trait features performs well, as biomass and diet directly encode functional roles.
> The motif-only FGW variant performs noticeably worse for large carnivores, demonstrating that simply inserting motif features into FGW does not reproduce the expressivity of our structural term. FGW-based backbones also exhibit structural inconsistencies:
>
> - **vanilla FGW:** connected but non-transitive backbones,
> - **FGW (traits as features, motifs as metric):** transitive but largely disconnected,
> - **FGW (motifs as both features and metric):** lower connectance than Wasserstein Motifs.
>
> Our method, in contrast, consistently produces backbones that are both connected and transitive and uniquely allows *controllable self-alignment* via the regularizer $\epsilon$, a capability FGW lacks.
>
> **On Partial-FGW.**
> We additionally examined Partial-FGW, as it is the closest FGW-based approach conceptually aligned with our goal of non-deterministic correspondences. However, we found it to be computationally infeasible at ecological scales. In an experiment computing pairwise alignments across a representative subset of networks (spanning the full range of 20–75 species), Partial-FGW required *up to 15 seconds per alignment* for networks with $\sim60$ nodes. In contrast, FGW and our method required up to $1$ second. Since our analysis requires $8256$ pairwise alignments, Partial-FGW would require several days of computation and is therefore not a practical baseline for ecological network data. This is consistent with prior reports of Partial-FGW scaling poorly when structural consistency dominates the cost.
>
> We have updated the manuscript to include these results (lines Appendix A.6). Overall, these findings demonstrate that our improvements stem from the *objective design*, not from feature choice, and that FGW/Partial-FGW baselines are neither structurally nor computationally comparable to our method.

---

> ### Author Response · Authors · 2025-11-22
> **Response to Reviewer j2Bj (2)**
>
> > The paper does not systematically study the behavior or provide robust defaults for the tradeoff parameter $\alpha$, self-alignment regularization parameter $\epsilon$ or step-size $\gamma$, nor how these interact.
>
> and
>
> > Also, could the authors do ablation study of the following: (1) first-order term off, (2) self-alignment penalty off?
>
> We thank the reviewer for highlighting the need to justify default hyperparameter choices and to clarify how they interact. Our revised manuscript now includes a systematic study of the model's behavior under variations of all three hyperparameters ($\alpha$, $\epsilon$, and $\gamma$), as summarized below.
>
> **Sensitivity and default value of $\alpha$**
> We set $\alpha = 0.5$ as the default to reflect an equal balance between the zeroth-order (feature-driven) and first-order (structure-driven) terms in Equation (1). This choice corresponds to the natural modelling assumption that motif-role similarity and structural similarity contribute comparably to ecological functional equivalence. Empirically, we found that the solution varies smoothly with $\alpha$: perturbing $\alpha$ by $\pm 0.1$ changes the alignment by only $\sim 0.12$ in the Frobenius norm. This small and approximately linear response demonstrates that (i) $\alpha = 0.5$ is a robust and interpretable default, and (ii) the method is not sensitive to moderate deviations from this value.
> Moreover, for a pair of food webs, we compute an alignment over several values of $\alpha$, noting the cost induced by the zeroth and first-order terms in each case.
> We observe stable behavior across $\alpha$, specifically that the zeroth-order and first-order costs remain practically constant, and the first-order cost consistently exceeds the zeroth-order cost.
> This further justifies $\alpha = 0.5$ as a reasonable trade-off between the respective costs.
> We have added these experiments to the updated manuscript (Appendix A.7 / Fig. 22).
>
> **Sensitivity and default value of $\boldsymbol{\epsilon}$.**
> We additionally examined how the total transported mass (controlled directly by $\epsilon$) varies with $\epsilon$. For a pair of food webs with sizes $m$ and $n$, respectively, the relationship follows a smooth saturating curve well-approximated by
> $$
> f(\epsilon) = a\bigl(1-\exp(-b\epsilon)\bigr),
> $$
> where $a \approx 0.83$ and $b\approx 4.25 (m+n) - 56.5$. This indicates that $\epsilon$ influences the degree of self-alignment in a *stable and predictable* manner. This also yields a practical scheme for selecting $\epsilon$ given a desired level of tolerated self-alignment. The updated manuscript includes a full sensitivity plot (Appendix A.7).
>
> **Sensitivity and default value of $\boldsymbol{\gamma}$.**
> In the problem formulation and algorithm, $\gamma$ effectively plays the role of a step-size in the usual sense of a first-order optimization algorithm.
> We found $\gamma = 0.1$ to be a suitable default value that strikes a balance between stability and the speed of convergence.
> To test this, we align several pairs of networks, using different values of $\gamma$, and show robustness to the choice of this parameter.
> Our theory shows convergence for any $\gamma > 0$ as shown in Theorem (1), thus the choice of gamma controls convergence rates, not convergence properties.
> In Fig. 24 of the updated manuscript, we present the results generated by our algorithm for *Cliff of Bandiagara* and *Namaqua* using various step sizes.
>
> **Ablation Analysis.**
> To further clarify the role of individual terms in the objective, we conducted two ablations:
>
> - **Self-alignment removed ($\epsilon=0$):** Performance deteriorates sharply. Meta-alignment consistency breaks down, and backbones fall below the null model in terms of transitivity, confirming that the self-alignment penalty is crucial in avoiding trivial, low-mass solutions.
> - **First-order term removed ($\alpha=0$):** Results remain comparable on most tasks, showing that the model is *robust* to the tradeoff parameter. Since $\alpha$ interpolates between two ecologically meaningful similarity sources (motif-based features and structural interactions), $\alpha=0$ represents a valid special case rather than a failure mode.
>
> These experiments, now incorporated into the manuscript (Appendix A.7), demonstrate that (i) our default hyperparameters are principled and empirically justified, (ii) the model responds smoothly to perturbations, and (iii) the method is stable and interpretable under variation or removal of individual terms.

---

> ### Author Response · Authors · 2025-11-22
> **Response to Reviewer j2Bj (3)**
>
> > If you keep the graph directed in the first-order term (e.g., separate in/out adjacency), do alignments/backbones improve?
>
> We thank the reviewer for the thoughtful question. Importantly, none of our theoretical results require the adjacency matrices $A_1$ and $A_2$ to be symmetric. The first-order term
> $$
> \langle T, A\_1 (T \odot C) A\_2\rangle
> $$
> remains well-defined for any nonnegative matrices, and directed adjacencies (e.g., separate in-/out-adjacency matrices) are therefore a drop-in replacement in our framework.
>
> However, using directed adjacencies changes the ecological interpretation of the first-order similarity. In this variant, the structural term compares only *directed neighbors* (e.g., prey sets), whereas our main formulation averages over predator--prey interactions by symmetrizing the adjacency. This shift places stronger weight on trophic directionality and weaker weight on undirected neighborhood structure.
>
> To evaluate this directed variant, we repeated all key experiments—pairwise alignments, role-consistency analysis, meta-alignment, and backbone quality—with directed adjacency matrices. While the method remains capable of distinguishing identical species from non-identical ones, we observed two consistent degradations:
>
> - **Meta-alignment performance declined substantially.** Alignment strengths across networks became noisier, and trophic-group structure was less pronounced.
> - **Backbone quality weakened.** Although the resulting top-$k$ backbones remained connected, their non-deterministic transitivity was only slightly above the null model, which is far below the improvement observed under our undirected formulation.
>
> Overall, while the framework technically supports directed adjacencies, our empirical analysis indicates that the directed version yields weaker alignment coherence and backbone structure. We attribute this degradation to the loss of structural information when only outgoing interactions (prey) are considered, whereas ecological role similarity is typically driven jointly by both predator and prey relationships. We included the relevant experiments in Appendix A.10.

---

> > ### Comment · Reviewer_j2Bj · 2025-11-27
> >
> > Thank you for the responses!

---

> > > ### Author Response · Authors · 2025-11-28
> > > **Thank for review, please consider revising your score**
> > >
> > > Thank you for your thoughtful engagement throughout the rebuttal. If you believe our responses adequately addressed your earlier concerns, we would be grateful if you would consider updating your evaluation. We sincerely appreciate your time and effort.

---

### Official Review · Reviewer_9DXb · 2025-11-01

**Soundness:** 2
**Presentation:** 3
**Contribution:** 2
**Rating:** 2
**Confidence:** 3

**Summary:**

This paper provides a formal mathematical framework for aligning ecological networks (food webs) using network motifs to identify structurally similar species and shared functional substructures. The authors show that existing motif-based approaches can be interpreted as minimizing a Fused Gromov-Wasserstein-like cost functional, which they call Wasserstein Motifs. Based on this formulation, they introduce an interpretable and theoretically grounded algorithm that efficiently computes non-deterministic (many-to-many) alignments between food webs represented as feature measure networks. The method also includes a procedure for identifying non-deterministic backbones of interactions, which capture functional redundancy among species.

**Strengths:**

- Captures many-to-many species relationships and functional redundancy
- Achieves computational speedup over previous motif-based approaches
- Identifies non-deterministic backbones revealing shared structural patterns across ecosystems

**Weaknesses:**

A central limitation is the absence of direct ecological ground truth. The evaluation assumes that identical species across networks have similar functional roles, an indirect validation that weakens confidence in the biological accuracy of the alignments and backbones. Without empirical or expert-labeled data, ecological correctness cannot be firmly established.

The dataset also contains inferred predator-prey interactions based on body size and taxonomy rather than direct field observations. While standard in metaweb studies, such inferred data can introduce bias and uncertainty, potentially affecting the reliability of the results.

Experimental design is limited by the use of fixed hyperparameters without sensitivity or ablation analysis, reducing confidence in the method's robustness and generalizability. Moreover, computing non-deterministic transitivity remains computationally demanding, restricting analysis to small backbones and limiting scalability.

No implementation code or supplementary materials are provided, which undermines reproducibility and prevents independent verification. Finally, the experimental results are largely qualitative and descriptive, lacking rigorous statistical evaluation, making it difficult to draw decisive or generalizable conclusions.

**Questions:**

Has the framework been tested or considered for other types of ecological or interaction networks? Demonstrating broader applicability could reinforce the method's general utility.

---

> ### Author Response · Authors · 2025-11-22
> **Response to Reviewer 9DXb (1)**
>
> > A central limitation is the absence of direct ecological ground truth. The evaluation assumes that identical species across networks have similar functional roles, an indirect validation that weakens confidence in the biological accuracy of the alignments and backbones. Without empirical or expert-labeled data, ecological correctness cannot be firmly established.
>
> We thank the reviewer for these thoughtful comments on ecological realism.
>
> Ecological ground truth and the “identical species” assumption.
> At continental scales and for many-site, multi-trophic food webs, it is currently infeasible to obtain standardized, expert-labeled “true” functional roles or interaction backbones for all species at all locations. This is a structural limitation of the available data, not a weakness specific to our work. As a result, large-scale ecological network studies routinely rely on indirect validation, for example, by checking consistency with traits, environmental gradients, or internal network statistics, rather than comparing to direct labels of roles.
>
> Our method does *not* assume during optimization that identical species across networks must have the same role. The non-deterministic transport plans and backbones are computed solely from network structure (and, for some baselines, traits), without using species identities as input. Species identity is used only *post hoc* as a diagnostic. A large body of work in food web and trait-based ecology demonstrates that species' functional roles are tightly linked to traits and tend to be conserved across communities when the same species occurs in multiple locations [7,8,9]. It is therefore reasonable to expect that any method that captures functional roles will, on average, assign more similar roles to the same species across different sites than to two arbitrary different species.
>
> We quantify this expectation in Fig. 8. We define an *alignment discrepancy* between two species as the average L1 distance between their alignment distributions across networks, and then compare this quantity for identical versus non-identical species. Identical species exhibit a mean discrepancy of $0.015 \pm 0.008$, which is substantially and statistically lower than the $0.024 \pm 0.007$ observed for non-identical species (lines 416-417).
> This difference is not imposed by construction and would not arise if the alignments were ecologically arbitrary.
> It provides a nontrivial, quantitative check that our alignments respect a key ecological regularity despite the absence of direct labels.
>
> We also deliberately use several additional, complementary validation steps so that our conclusions do not hinge on a single heuristic:
> (i) our alignments largely respect coarse dietary classes (carnivore, omnivore, herbivore) and trait-based species groups, and clear mismatches that appear in baselines (for instance, aligning a wildcat to a gazelle) are avoided by our approach (Fig. 5, lines 36–45);
> (ii) meta-alignments aggregated by species groups display strong within-group alignment for small and large carnivores and herbivores (Fig. 10), consistent with their known trophic roles; and
> (iii) non-deterministic backbones inferred from our alignments exhibit systematically higher relative connectance, connectivity, and transitivity than random subnetworks of the same size (Fig. 12 and Def. 4.3), following the established criteria of [10].

---

> ### Author Response · Authors · 2025-11-22
> **Response to Reviewer 9DXb (2)**
>
> > The dataset also contains inferred predator-prey interactions based on body size and taxonomy rather than direct field observations. While standard in metaweb studies, such inferred data can introduce bias and uncertainty, potentially affecting the reliability of the results.
>
> We also acknowledge that the food webs are derived from a regional metaweb that combines documented and inferred interactions. This construction, however, is identical to the one used in the current state-of-the-art dataset for Sub-Saharan African mammal food webs [10, 11]  and reflects the standard methodology in macroecological network analysis.
>
> A comprehensive binary metaweb is constructed from the union of predator–prey interactions in *Mammals of Africa* [12] and a global mammal interaction database [11]. This metaweb is then supplemented with additional predator–prey links inferred from taxonomy and predator–prey body size overlap [8, 9, 13, 14]. Local food webs are obtained by sub-sampling this metaweb for the species present at each site, following the widely used metaweb framework for large-scale food web comparison [15, 16]. Subsequent ecological work that relies on the same dataset has made clear that the resulting metaweb encodes *potential* interactions: when two species co-occur, the metaweb records that an interaction can occur, but not that it always occurs or with a specific strength.
>
> The use of body size and taxonomy to infer additional interactions is strongly grounded in ecological theory and data. Niche-like models, in which predators consume a continuous range of prey ordered by body size, reproduce empirical food web structures with high accuracy, including degree distributions, trophic levels, and motif frequencies [7, 17]. More recent syntheses confirm that predator traits, particularly body size, are among the strongest predictors of who eats whom and of overall food web architecture across ecosystems [8, 9, 10, 19]. These studies show that mammal food webs constructed from this metaweb approach exhibit realistic structural properties at continental scales. Thus, although uncertainty at the level of individual links is unavoidable, the global structural patterns that our method exploits are supported by a substantial body of empirical and theoretical work.
>
> From the perspective of machine learning, our contribution is purposefully orthogonal to the details of network construction. We take the food webs as given input graphs and introduce a new optimal transport motif alignment and non-deterministic backbone framework. All methods compared in our experiments, including baselines, operate on the same metaweb-derived networks. Any bias introduced by inferred interactions, therefore, affects all methods equally, and our empirical claims concern relative improvements in alignment quality and backbone coherence under identical data assumptions. At the same time, the framework itself is agnostic to how the networks are built. If richer information becomes available, such as weighted or temporal interaction strengths, it could be incorporated without changing the core methodology.

---

> ### Author Response · Authors · 2025-11-22
> **Response to Reviewer 9DXb (3)**
>
> > Experimental design is limited by the use of fixed hyperparameters without sensitivity or ablation analysis, reducing confidence in the method's robustness and generalizability.
>
> We thank the reviewer for pointing out the need for hyperparameter sensitivity and ablation analyses. In response, we conducted a series of additional experiments to directly address this concern and show that our model is *robust* with respect to both hyperparameters ($\alpha$ and $\epsilon$) and clarify the functional role of each term in the objective.
>
> **Sensitivity Analysis.** We examined how the total transported mass (the key quantity controlled by $\epsilon$ in Equation (1)) varies as $\epsilon$ changes. For a pair of food webs with sizes $m$ and $n$, respectively, the relationship follows a smooth saturating curve well-approximated by
> $$
> f(\epsilon) = a(1-\exp(-b \epsilon)),
> $$
> where $a \approx 0.83$ and $b\approx 4.25 (m+n) - 56.5$. This indicates that $\epsilon$ influences the solution in a stable and predictable manner. This fit also provides a *practical default scheme* to choose $\epsilon$ given a desired level of self-alignment penalty.
>
> For $\alpha$, we perturbed its values locally around $\alpha_0 = 0.5$. The Frobenius distance between the resulting alignments was approximately linear in $\alpha$, and a $\pm 0.1$ perturbation in $\alpha$ changed the alignment by only $\sim0.12$ in the Frobenius norm. This small and smoothly varying response shows that the method is insensitive to moderate changes in $\alpha$.
>
> We have updated our manuscript to include both hyperparameter sensitivity experiments, which are presented in Appendix A.7.
>
> **Ablation Analysis.** We further evaluated the role of each hyperparameter by running all alignment experiments under two ablations: (1) $\epsilon = 0$ (removing the self-alignment penalty) and (2) $\alpha = 0$ (removing the first-order structural term). For both settings, we recomputed alignment discrepancy (Fig. 8), meta-alignment (Fig. 9), and backbone quality metrics (Fig. 12)
>   - **Self-alignment ablation ($\epsilon = 0$):** Performance degrades substantially, consistent with theory. Without the penalty, the objective allows for trivial, low-mass alignments, resulting in collapsed meta-alignments and backbones with transitivity scores lower than those of the null model. This confirms that the self-alignment term is essential.
>   - **Tradeoff ablation ($\alpha = 0$):** Results remain comparable to the default model, indicating that the method is robust with respect to the tradeoff parameter. Since $\alpha$ interpolates between two valid sources of similarity (features and structure), $\alpha$ = 0 is a meaningful special case rather than a failure mode.
>
> We have also updated our manuscript to include these ablation experiments, which are presented in Appendix A.7.
>
> Taken together, these experiments demonstrate that the method’s behavior is stable, interpretable, and robust under variation or removal of individual hyperparameters.
>
> > Moreover, computing non-deterministic transitivity remains computationally demanding, restricting analysis to small backbones and limiting scalability.
>
> We agree with the reviewer that computing non-deterministic transitivity is computationally demanding. Importantly, however, *this computation is not part of the alignment algorithm itself*. It is used only once, as an *evaluation metric* for assessing backbone quality in Section 4, and is *not required for any user of our framework*. The alignment and backbone extraction steps remain fast and scalable (0.05 seconds per alignment across 8,256 pairs).
>
> To address the reviewer’s concern more precisely, we performed a theoretical analysis of the transitivity computation given a dataset of $N$ food webs and a backbone size $k$. Definition 4.3 involves summing over all triples of networks and all triples of backbone species, yielding time complexity
> $$
> O(N^3k^3),
> $$
> which is polynomial time and tractable for typical ecological datasets. In practice, the transitivity computation is dominated by the *null-model estimation*, which requires recomputing the metric across 50 random subsets. On our mid-range desktop CPU (AMD Ryzen 7 9700X), the entire null-model computation took approximately 20 hours, a cost incurred once solely for evaluation purposes.
>
> We have clarified this distinction in the revised manuscript (line 507-509).

---

> ### Author Response · Authors · 2025-11-22
> **Response to Reviewer 9DXb (4)**
>
> > No implementation code or supplementary materials are provided, which undermines reproducibility and prevents independent verification.
>
> We appreciate the reviewer’s concern regarding reproducibility. We clarify that the supplementary materials were already included in the original submission, which contains algorithmic details, proofs, visualizations, and species-level correspondences (pages 13-20 of the original submission).
> In the revised manuscript, we have further expanded these supplementary materials with additional theoretical results and new experimental analysis (in blue). Thus, all derivations and evaluation procedures have been available to the reviewers from the beginning.
>
> Since the Sub-Saharan African food-web dataset used in our experiments is not publicly available, the released repository will focus on the method rather than the dataset. To ensure that the community can still run and verify the Wasserstein Motifs framework end-to-end, we additionally provide code for generating synthetic ecological networks using the classical niche model [7] and cascade model [20]. These generators enable the reproduction of all algorithmic components, including alignment, backbone computation, and backbone evaluation, on synthetic ecological networks. The anonymized repository can be found at [https://anonymous.4open.science/r/Wasserstein-Motifs-Synthetic-46C3](https://anonymous.4open.science/r/Wasserstein-Motifs-Synthetic-46C3).
>
> >Finally, the experimental results are largely qualitative and descriptive, lacking rigorous statistical evaluation, making it difficult to draw decisive or generalizable conclusions.
>
> We appreciate this concern and agree that rigorous quantitative evaluation is essential. Our intention was to combine interpretable ecological case studies with quantitative analyses across all 129 food webs. We recognize that this point may not have been sufficiently emphasized, and we will clarify and strengthen the statistical aspects as discussed below.
>
> Beyond the illustrative examples in Fig. 5, our main ecological claims are supported by several quantitative analyses:
>
> **Species level alignment consistency (Fig. 8).**
> As mentioned in our first response, we define an *alignment discrepancy* between two species as the average L1 distance between their alignment distributions across networks, and we compare this quantity for identical versus non-identical species. It provides a quantitative, distribution-level validity check that our method assigns more consistent roles to the same species across sites than to unrelated species, which is a central ecological expectation.
>
> **Dataset scale and replication.** All alignment and backbone analyses are performed on a continental-scale dataset of 129 mammal food webs that cover a wide range of species richness (from 16 to 75 species) and connectance (from 0.0345 to 0.221), and encompass 216 species from 12 orders and 33 families (Fig. 6). This large number of independent networks provides substantial replication across ecological contexts, which supports the generality of the patterns we report, in line with standard practice in macroecological network analysis.
>
> **Backbone evaluation against a null model (Fig. 12).** For each backbone size $k$, we compute, across all networks, the median relative connectance, the path likelihood, and the non-deterministic transitivity score (Def. 4.3). We then compare these quantities to a null model in which the same statistics are computed on subnetworks induced by $k$ species selected uniformly at random, averaged over 50 trials per network and per $k$. Our backbones show systematically higher relative connectance and higher probability of being connected, and they are consistently more transitive than the null model for the range of $k$. This follows the null model-based evaluation framework introduced by [21] for ecological backbones and by many subsequent ecological network studies that rely on randomization tests rather than explicit parametric models.
>
> We compare against strong optimal transport baselines such as FGW and Partial FGW, under the same data and feature assumptions, and we highlight not only qualitative differences (for example, ecologically implausible carnivore–herbivore alignments produced by some baselines) but also global patterns in alignment consistency, group-level meta-alignments (Fig. 9), and backbone statistics (Fig. 12). Thus, our conclusions about improved alignment quality and backbone structure are based on systematic differences across methods, not only on isolated examples.
>
> Taken together, these elements quantify the consistency of functional role across species levels and benchmark backbone structure against an explicit null model in 129 independent networks.

---

> ### Author Response · Authors · 2025-11-22
> **Response to Reviewer 9DXb (5)**
>
> > Has the framework been tested or considered for other types of ecological or interaction networks? Demonstrating broader applicability could reinforce the method's general utility.
>
> We thank the reviewer for this question. While our work focuses on ecological networks, the proposed framework is *inherently general*. The formulation in Eq. (1) aligns two featured measure networks, a class that includes any network endowed with (i) node features, (ii) a finite measure, and (iii) adjacency structure. Motif-role profiles serve as domain-appropriate features for food webs, but the objective and KL-BAPG solver operate independently of ecological semantics.
>
> To demonstrate broader applicability to ecological networks, we also evaluated our method on a subset of food webs from [21], for which we were able to obtain the directed interaction data. However, the dataset does not include species-level trait profiles or other ecological attributes that would allow for quantitative ecological validation.
>
> To further demonstrate the general utility, we also aligned synthetic ecological networks generated by the classical Niche model [7] and the Cascade model [20]. These synthetic networks differ structurally from empirical food webs, yet the Wasserstein Motifs framework produced stable and interpretable non-deterministic alignments in both settings.
>
> Examples of these alignments are now included in the updated manuscript (Appendix A.8).
>
> Taken together, these empirical and synthetic experiments show that the proposed approach applies robustly across multiple ecological network types and is not restricted to a single dataset or specific empirical setting.

---

> > ### Comment · Reviewer_9DXb · 2025-11-26
> >
> > Thank you to the authors for the rebuttal. It addresses some of my earlier concerns and provides helpful clarification, so I am making an upward adjustment to my score.

---

> > > ### Author Response · Authors · 2025-11-28
> > > **Thank you for review, please consider revising your score**
> > >
> > > We thank you again for the constructive feedback during the rebuttal and for updating the score. To further improve the paper, we would appreciate it if you could let us know which concerns remain inadequately addressed. We believe we responded thoroughly to each of the questions and weaknesses you raised, and we summarize the main revisions below:
> > >
> > >  **Clarifications added directly in the rebuttal (not reflected as new experiments):**
> > >
> > > - Ecological realism/lack of ground truth: Explained the standard reliance on indirect validation in large-scale food-web studies and why species identity is used only post hoc.
> > > - Inferred interactions in metawebs: Clarified that the dataset construction follows standard macroecological methodology and affects all baselines equally.
> > > - Qualitative vs. quantitative evaluation: Highlighted the quantitative elements already present (alignment discrepancy statistics, null-model backbone tests across 129 networks).
> > >
> > > **Full hyperparameter sensitivity analysis (Appendix A.7, Figs. 22–24)**
> > >
> > > - Characterized the $\epsilon$–total-mass relationship with a saturating curve fit ($R^2$ = 0.984).
> > > - Demonstrated smooth dependence on α (±0.1 perturbation led to a $\approx$0.12 Frobenius deviation).
> > > - Verified robustness to the choice of step size $\gamma$.
> > >
> > > **Comprehensive ablation studies (Appendix A.7, Figs. 25–28)**
> > > Recomputed discrepancies, meta-alignments, and backbone statistics across all 8,256 alignment pairs:
> > >
> > > - Removed first-order term ($\alpha=1$).
> > > - Removed self-alignment penalty ($\epsilon=0$).
> > >
> > > These confirm (i) the necessity of the self-alignment penalty and (ii) robustness with respect to $\alpha$.
> > >
> > > **Broader applicability experiments (Appendix A.8)**
> > > - Added alignments on three additional empirical networks.
> > > - Added alignments on six synthetic networks generated via the Niche and Cascade models.
> > >
> > > **Reproducibility enhancements**
> > > - Released anonymized code with Niche/Cascade generators for full end-to-end reproducibility despite the dataset being non-public:
> > > https://anonymous.4open.science/r/Wasserstein-Motifs-Synthetic-46C3
> > >
> > > If you have any particular concerns or aspects that require further clarification, we would sincerely appreciate your guidance. Your feedback will help us strengthen the final version.

---

### Official Review · Reviewer_gwYn · 2025-11-01

**Soundness:** 2
**Presentation:** 1
**Contribution:** 3
**Rating:** 4
**Confidence:** 3

**Summary:**

The paper studies the problem of identifying similarities in different ecosytems by formulating it as a network alignment problem. Given a directed graph, where nodes are species and edges as predation, the goal is to identify species across different ecosystems (nodes across graphs) that play functionally similar roles. The authors propose an objective function that captures node features, graph structure and a regularizing term to disallow self-alignments (node mapped to itself). They then provide an algorithm that minimizes this objective and prove convergence to stationary point. Experiments are provided on real world data to show that the alignments obtained by their methods are faster and more coherent.

**Strengths:**

The problem considered and the challenges in solving it are well motivated. The main contributions are the formalization of alignment problem and showing that the heuristic approach used widely in the field, is a special case for their formulation. The algorithmic approach seems to be relied mostly on previous work, but they show that optimizing for alignment under their cost results in more coherent results. As there is no ground-truth, the evaluations are conceptual. The non-deterministic transitivity score is well specified and also consistent with their goal.

**Weaknesses:**

The theoretical claims may be seen as a bit of oversell. The convergence is proven under an assumption that is not justified well or stated under which conditions might hold, but instead some empirical evidence is provided that this holds. This is a bit non-standard for theoretical proofs. In the comparison against baselines for runtime, is motif computation, enumeration considered? For methods with and without that might be a big difference. Also, the $L1$ alignment distribution is not reported for the baselines.

**Questions:**

The matrix $C$ is assumed to have positive entries, but with $\varepsilon=1$ and correlation as the dissimilarity, the entries could be negative. Was some other dissimilarity measure used for part of experiments?

---

> ### Author Response · Authors · 2025-11-22
> **Response to Reviewer gwYn (1)**
>
> > The theoretical claims may be seen as a bit of oversell. The convergence is proven under an assumption that is not justified well or stated under which conditions might hold, but instead some empirical evidence is provided that this holds. This is a bit non-standard for theoretical proofs.
>
> Thank you for pointing this out. This is a running assumption in the state-of-the-art analysis for Bregman Proximal Algorithms [1,2,3,4,5]. Recent literature consistently indicates that this is an open, challenging, and non-trivial problem [6,Prop. 3.5].  We understand that this is not a justification for not addressing the issue, but rather provides context for our assumption and the empirical verification that supports it. We want to be transparent about it. Nonetheless, in a new Appendix, we show that under additional conditions, such as sufficiently small and decaying step sizes, we can guarantee that Assum. 1 holds for our proposed algorithm. This is a brand new analysis of general independent interest.
>
> **Why Assum. 1 is natural and not ad hoc**
>
> Our algorithm uses the entropic generator $h(X)=\sum_{ij}x_{ij}\log x_{ij}$, whose Bregman divergence $D_h$ is non-symmetric. Because the two blocks (row/column updates) are also applied in a non-symmetric fashion, the usual Lyapunov argument produces at each iteration an additional *asymmetric* term
> $$
>   D_h(\hat T^{(k)},T^{(k+1)}) - D_h(T^{(k+1)},\hat T^{(k)}),
> $$
> which does not cancel. Assum. 1 is precisely the requirement that the sum of these asymmetric terms be absolutely convergent. This is exactly the standard "summable error'' condition that appears in inexact proximal and KL-based convergence analyses.
>
> The *same* type of condition is used explicitly in closely related work. [6] introduces a Bregman Alternating Projected Gradient method and assumes a bounded *accumulative asymmetric error of the Bregman distance* in their convergence theorem. They explicitly remark that this condition is easy to verify in the quadratic case, but technically challenging in the entropic case; thus, they provide empirical evidence to support the condition. In the quadratic case, the asymmetry vanishes identically; in the entropic case, one faces exactly the same technical difficulty, and we adopt exactly the same kind of assumption.
>
> Assum. 1 is aligned with the classical "summable error'' paradigm in inexact proximal methods. [1]  already assumes that the sequence of inexactness errors is absolutely summable to guarantee convergence of inexact proximal point schemes. [2] develops a general KL-based descent framework that allows relative error tolerances and recovers convergence under suitable control of the accumulated error terms. In the Bregman setting,  [3,4] both employ inexact Bregman proximal schemes in which convergence is obtained under conditions of the form $\sum_k \varepsilon_k < \infty$ on the inexactness/error sequence. More recently, [5] extends this inexact Bregman framework to DC problems with similar summable or controlled error
> criteria.
>
>
> **New sufficient conditions under which Assum. 1 is *proved* for entropy.**
>
> To address the reviewer’s concern more constructively, we have added a new appendix that demonstrates how Assum. 1 is not merely postulated, but can be rigorously verified for the proposed entropic algorithm under mild additional assumptions (standard in optimization and compatible with our setting).
>
> Concretely, in Appendix C we prove the following:
>  - Using a third-order Taylor expansion of the scalar KL divergence, we establish a local cubic
>   bound on the asymmetry: for $u,v>0$ with $|u-v|\le \tfrac12\min(u,v)$,
>  $$
>     |D(u\|v) - D(v\|u)|
>     \le
>     C_0 \frac{|u-v|^3}{\min(u,v)^2},
>  $$
>   with explicit constant $C_0=2$. Applied entry-wise to matrices, this bounds the asymmetric part of
>   $D_h$.
>
> - We then prove a "marginal stability'' lemma along the actual iterates (Lemma C.3): because the cost operators $C_\varepsilon,A_1,A_2$ are bounded, and the KL projections keep the iterates in
>   the feasible sets $C_1(\mu)$ and $C_2(\nu)$, the second (column) block satisfies
> $$
>     |T^{(k+1)}\_{ij} - \hat T^{(k)}\_{ij}\|
>     \le
>     C_g\gamma_k\min(T^{(k+1)}\_{ij},\hat T^{(k)}\_{ij}),
> $$
>   with an explicit constant $C_g = 40 L_Q$, where $L_Q$ is a uniform bound on the gradient terms
>   $Q^{(k)},Q'^{(k)}$ (Lemma C.1).
>
> Combining these two ingredients, we show (Theorem C.4) that the per-iteration asymmetric error is
> bounded by
> $$
>   \bigl|D_h(\hat T^{(k)},T^{(k+1)}) - D_h(T^{(k+1)},\hat T^{(k)})\bigr|
>   \le C_*\gamma_k^3,
> $$
> for an explicit constant $C_*>0$ depending only on $\mu,\nu,A_1,A_2,C_\varepsilon$. Therefore, under the standard summability condition $\sum_{k=0}^\infty \gamma_k^3 < \infty$ (a usual requirement for decaying step-size schemes), we obtain
> $$
>   \sum_{k=0}^\infty
>     \bigl|D_h(\hat T^{(k)},T^{(k+1)}) - D_h(T^{(k+1)},\hat T^{(k)})\bigr|
>   < \infty,
> $$
> i.e., Assum. 1 holds automatically for our entropic algorithm with decaying step sizes.

---

> ### Author Response · Authors · 2025-11-22
> **Response to Reviewer gwYn (2)**
>
> > In the comparison against baselines for runtime, is motif computation, enumeration considered? For methods with and without that might be a big difference.
>
> We thank the reviewer for pointing out this detail. The runtimes reported in Section 3 (lines 405-407 in the revised manuscript) are recorded after *precomputed cost matrices*. To clarify the impact of motif enumeration, we conducted additional experiments in which the motif profiles and cost matrices were recomputed for each pair of food webs in the Sub-Saharan Africa dataset. These experiments show that *about $75$% ($0.15\pm0.07$ seconds out of $0.20\pm0.08$ seconds) of the total computation time is spent on cost-matrix construction*, which includes motif enumeration. Thus, when a user aligns *a single pair* of food webs, motif computation is indeed the dominant cost, but the total runtime remains small (typically less than $1$ second on commercial laptops for large food webs with up to $60$ species), and the alignment step itself remains lightweight.
> We have clarified this point in the updated manuscript (lines 400-403).
>
> More importantly, this overhead occurs *only once*. For any dataset-level application involving many pairwise alignments (e.g., backbone computation), motif-role profiles can be computed *a priori for each graph*. In that setting, computing a cost matrix reduces to *simple pairwise dissimilarity of the precomputed motif-role profiles*, eliminating motif enumeration entirely from the runtime budget and restoring the alignment solver as the primary computational bottleneck. This also means that users can freely sweep hyperparameters such as $\alpha, \epsilon$, or $\gamma$ without ever recomputing motif counts. We have clarified this distinction in the revision (lines 403-405).
>
> > Also, the L1 alignment distribution is not reported for the baselines.
>
> We thank the reviewer for pointing out the missing comparison in the alignment discrepancy experiment (Fig. 8). In response, we computed the same discrepancy metric for all baselines and variants (FGW; FGW with traits and motifs; FGW with only motifs; Wasserstein Motifs with directed adjacencies) and updated the manuscript to include them (Appendix A.6). Across all methods, we observed very similar trends: identical species consistently exhibit lower L1 discrepancy than non-identical species. These results indicate that all baseline methods produce ecologically sensible alignments in this indirect validation task. This also supports our interpretation that the experiment serves as a *validity check*: it verifies that a method can capture the common assumption that identical species play more similar roles across food webs, but it does not provide sufficient discriminative power to separate models with different formulations.
>
> We emphasize that our formulation allows for self-alignment, meaning that the alignment distribution of a species does not necessarily sum to one. In contrast, FGW-based baselines enforce mass conservation, so every alignment distribution must be a probability distribution. As a result, absolute differences in the alignment discrepancy values partly reflect the difference in the feasibility region of the transport plans, not just model performance. For these reasons, we report only the qualitative trend (of alignment discrepancies of identical vs. non-identical species) rather than comparing absolute magnitudes across methods.
>
> > The matrix C is assumed to have positive entries, but with $\epsilon = 1$ and correlation as the dissimilarity, the entries could be negative. Was some other dissimilarity measure used for part of experiments?
>
> We appreciate the reviewer's attention to the distinction between the *base* cost matrix $C$ and the *shifted* cost matrix $C_\epsilon$ used in the objective functional. To clarify: *the feature dissimilarity matrix $C$* is always entry-wise non-negative, since it is constructed from a non-negative discrepancy $d(\cdot, \cdot) \geq 0$. The apparent negativity arises only in the *shifted cost matrix $C_\epsilon$*
> $$
> C_\epsilon := C - \epsilon 1_m1_n^\top,
> $$
> introduced solely for notational simplicity and does not alter the underlying dissimilarity used to compute $C$.  We have added a clarifying paragraph in the updated manuscript to make this explicit (lines 170-173).

---

> ### Author Response · Authors · 2025-11-22
> **Additional Comments for Reviewer gwYn**
>
> We also noticed that reviewer gwYn assigned a "$1$: poor" for presentation quality. We appreciate the reviewer's effort in evaluating our work; however, none of the written comments appear to reference issues related to clarity or organization. To help us improve the presentation of the paper, we would be very grateful if the reviewer could share specific concerns or concrete suggestions regarding the clarity or structure of the manuscript. In response to this score, we have already taken steps to improve the presentation. Most notably, we added a *pipeline diagram* (Appendix A.9) summarizing the Wasserstein Motif workflow, from raw ecological networks to alignment, backbone computation, and other downstream ecological analyses. Any further guidance would be greatly appreciated.

---

### Author Response · Authors · 2025-11-22
**Summary of Responses**

We thank the three reviewers for their comments, questions, and insights. We will respond to each reviewer individually, addressing their questions. Nonetheless, we would like to summarize the set of changes we have made and responses to the issues they have raised. These responses have been incorporated into the manuscript as well, including new text and appendices with new experiments, all of which are marked in blue. We believe these are comprehensive, and would appreciate a revision of the scores the reviewers have assigned to our paper. If there are unsatisfactory answers and issues yet to be addressed, we will be ready to work on them.

- **Assumption 1 and convergence.**
We clarified that Assumption 1 is a standard summable-error condition in Bregman proximal methods and added Appendix C, where we prove that, for entropic KL and decaying step sizes with $\sum_k \gamma_k^3 < \infty$, Assumption 1 holds and convergence follows.

- **Hyperparameters and ablations ($\alpha,\epsilon,\gamma$).**
We performed sensitivity analyses showing smooth, modest dependence on $\alpha$ and a predictable saturating effect of $\epsilon$ on transported mass, justified $\alpha=0.5$ as a robust default, demonstrated robustness to $\gamma$, and showed via ablations that the self-alignment term is crucial while the method is stable to removing the first-order term.

- **Runtime, motifs, and transitivity.**
We clarified that reported runtimes assume precomputed costs, quantified that $\sim 75$% of per-pair time is spent in one-time motif/cost construction, and stressed that non-deterministic transitivity (with $O(N^3 k^3)$ complexity) is used only once as an offline evaluation metric, not in routine use.

- **Ecological realism and inferred interactions.**
We emphasized that species identities are used only for post hoc validation, not in the optimization, that our indirect checks (identical vs. non-identical species, trophic groups, null-model backbones) follow standard macroecological practice, and that metaweb-based, trait/body-size inferred links are standard and affect all methods equally.

- **Relation to FGW and fair baselines.**
We made explicit that our edge-filtered, feature-modulated quadratic term cannot be written as FGW with any metrics/loss, and added FGW variants with the same motif features, showing that they reproduce some trends but generally yield weaker, less connected or less transitive backbones than our method; Partial-FGW is empirically too slow at our scale.

- **Additional experiments (L1, directed graphs, other networks).**
We now report L1 alignment discrepancy for all methods, tested a directed-adjacency variant (which produced weaker meta-alignments and backbones than the undirected version), and demonstrated applicability on additional empirical food webs and synthetic niche/cascade networks.

- **Clarifications, presentation, and reproducibility.**
We clarified the roles of $C$ vs. $C_\epsilon$, expanded and updated the supplemental material with new theory and experiments, added a pipeline diagram to improve presentation, and released anonymized code with synthetic-network generators to ensure end-to-end reproducibility despite the non-public dataset.

---

> ### Author Response · Authors · 2025-11-22
> **References**
>
> [1] Osman Guler. New proximal point algorithms for convex minimization. SIAM Journal on Optimization, 2(4):649–664, 1992.
>
> [2] Hedy Attouch, Jerome Bolte, and Benar Fux Svaiter. Convergence of descent methods for semi-algebraic and tame problems: proximal algorithms, forward–backward splitting, and regularized gauss–seidel methods. Mathematical programming, 137(1):91–129, 2013.
>
> [3] Lei Yang and Kim-Chuan Toh. Bregman proximal point algorithm revisited: A new inexact version and its inertial variant. SIAM Journal on Optimization, 32(3):1523–1554, 2022.
>
> [4] Hong TM Chu, Ling Liang, Kim-Chuan Toh, and Lei Yang. An efficient implementable inexact entropic proximal point algorithm for a class of linear programming problems. Computational Optimization and Applications, 85(1):107–146, 2023.
>
> [5] Lei Yang, Jingjing Hu, and Kim-Chuan Toh. An inexact bregman proximal difference-of-convex algorithm with two types of relative stopping criteria. Journal of Scientific Computing, 103(3):91, 2025.
>
> [6] Jiajin Li, Jianheng Tang, Lemin Kong, Huikang Liu, Jia Li, Anthony Man-Cho So, and Jose Blanchet. A convergent single-loop algorithm for relaxation of gromov-wasserstein in graph data. arXiv preprint arXiv:2303.06595, 2023.
>
> [7] Richard J Williams and Neo D Martinez. Simple rules yield complex food webs. Nature, 404(6774):180–183,
> 2000.
>
> [8] Dominique Gravel, Timothee Poisot, Camille Albouy, Laure Velez, and David Mouillot. Inferring food web structure from predator–prey body size relationships. Methods in Ecology and Evolution, 4(11):1083–1090, 2013.
>
> [9] Ulrich Brose, Phillippe Archambault, Andrew D Barnes, Louis-Felix Bersier, Thomas Boy, Joao Canning-Clode, Erminia Conti, Marta Dias, Christoph Digel, Awantha Dissanayake, et al. Predator traits determine food-web architecture across ecosystems. Nature ecology & evolution, 3(6):919–927, 2019.
>
> [10] Bernat Bramon Mora, Alyssa R Cirtwill, and Daniel B Stouffer. pymfinder: a tool for the motif analysis of binary and quantitative complex networks. BioRxiv, page 364703, 2018.
>
> [11] Kai M. Hung, Lydia Beaudrot, Ann E. Finneran, Alex G. Zalles, and Cesar A. Uribe. Quantifying functionally equivalent species and ecological network dissimilarity with optimal transport distances. Methods in Ecology and Evolution, n/a(n/a), 2025.
>
> [12] Evan C Fricke, Chia Hsieh, Owen Middleton, Daniel Gorczynski, Caroline D Cappello, Oscar Sanisidro, John Rowan, Jens-Christian Svenning, and Lydia Beaudrot. Collapse of terrestrial mammal food webs since the late pleistocene. Science, 377(6609):1008–1011, 2022.
>
> [13] Jonathan Kingdon. Mammals of Africa. Bloomsbury, 2013.
>
> [14] Simon T Segar, Tom M Fayle, Diane S Srivastava, Thomas M Lewinsohn, Owen T Lewis, Vojtech Novotny, Roger L Kitching, and Sarah C Maunsell. The role of evolution in shaping ecological networks. Trends in Ecology & Evolution, 35(5):454–466, 2020.
>
> [15] Guy Woodward, Dougie C Speirs, Alan G Hildrew, and C Hal. Quantification and resolution of a complex,size-structured food web. Advances in ecological research, 36:85–135, 2005.
>
> [16] Mercedes Pascual and Jennifer A Dunne. Ecological networks: linking structure to dynamics in food webs. Oxford University Press, 2005.
>
> [17] Loıc Pellissier, Camille Albouy, Jordi Bascompte, Nina Farwig, Catherine Graham, Michel Loreau, Maria Alejandra Maglianesi, Carlos J Melian, Camille Pitteloud, Tomas Roslin, et al. Comparing species interaction networks along environmental gradients. Biological reviews, 93(2):785–800, 2018.
>
> [18] Daniel B Stouffer, Enrico L Rezende, and Luıs A Nunes Amaral. The role of body mass in diet contiguity and
> food-web structure. Journal of Animal Ecology, 80(3):632–639, 2011.
>
> [19] Lydia Beaudrot, Miguel A Acevedo, Daniel Gorczynski, and Nyeema C Harris. Geographic differences in body size distributions underlie food web connectance of tropical forest mammals. Scientific Reports, 14(1):6965, 2024.
>
> [20] Joel E Cohen and Charles M Newman. A stochastic theory of community food webs i. models and aggregated data. Proceedings of the Royal society of London. Series B. Biological sciences, 224(1237):421–448, 1985.
>
> [21] Bernat Mora, Dominique Gravel, Luis J Gilarranz, Timothee Poisot, and Daniel B Stouffer. Identifying a common backbone of interactions underlying food webs from different ecosystems. Nature Communications, 9(1):2603,2018

---

### Meta-Review · Area_Chair_tYSf · 2026-01-03

**Summary:**

The paper proposes an optimal-transport-inspired framework, termed Wasserstein Motifs, for aligning ecological food-web networks in order to identify functionally equivalent species across ecosystems, introducing a motif-based representation, a non-deterministic alignment algorithm, and large-scale experiments on 129 mammal food webs to extract shared interaction backbones. While the topic is interesting and the combination of OT ideas with ecological network motifs is original in spirit, several major weaknesses remain even after a substantial rebuttal. First, the theoretical guarantees remain somewhat fragile: although the authors strengthened the convergence analysis by clarifying that their key assumption follows from standard Bregman proximal arguments and by adding sufficient conditions under which it holds in the entropic KL setting, the result still relies on additional constraints that may not cover all practical regimes. Second, the empirical evaluation, while broadened and clarified in the rebuttal, still raises interpretability and comparability concerns: runtime reporting depends on a separation between preprocessing and alignment that may not hold in all use cases, baseline comparisons require careful normalization to be fully convincing, and some modeling choices (such as undirected handling of inherently directed food webs) remain only partially justified. Third, the biological validation, despite thoughtful indirect checks and added analyses, necessarily relies on inferred interaction data and lacks direct ecological ground truth; while this limitation is standard in macroecology and reasonably acknowledged, it constrains the strength of the biological conclusions. Finally, although many methodological concerns (feature fairness, sensitivity to hyperparameters, ablations, and reproducibility) were addressed through additional experiments, code release, and feature-matched baselines, the scope and number of post-rebuttal changes underscore that the current presentation still requires deeper consolidation. In conclusion, the authors clearly made a major rebuttal effort and improved the work along several critical dimensions, but the remaining theoretical caveats and empirical ambiguities suggest that an in-depth rewrite and tighter integration of assumptions, methodology, and validation are needed, and the paper cannot be accepted in its current form.

**Reviewer Concerns:**

Several substantive reviewer concerns were meaningfully addressed in the rebuttal. In particular, questions about feature fairness and comparisons to FGW-style baselines were handled through feature-matched baselines, additional discrepancy metrics, and clearer positioning relative to FGW and Partial-FGW. Concerns about hyperparameter robustness and ablations were also addressed with extensive sensitivity analyses and targeted ablation studies that clarified the role of each term in the objective. Reproducibility issues were mitigated by releasing anonymized code and synthetic data generators. However, other concerns remain only partially resolved. The convergence analysis, while strengthened, still depends on assumptions whose practical scope is unclear, leaving some reviewers’ doubts intact. Likewise, issues related to biological validation without ground truth and to modeling choices such as undirected graph handling persist as structural limitations rather than resolved points. Finally, the runtime and scalability discussion, though clarified, still relies on assumptions about preprocessing reuse that may not generalize across applications.

**Reviewer Scores:**

- gwYn (initial score: 4): Possible increase, reflecting satisfaction with the clarified convergence analysis, added ablations, and improved runtime accounting, while still retaining some reservations about theoretical assumptions.
- 9DXb (initial score: 2): Mention increasing during discussion; with full participation, this reviewer would likely remain cautious, possibly settling around 4, acknowledging the added experiments and clarifications but still viewing the lack of direct ecological validation as a fundamental limitation.
- j2Bj (initial score: 6): Likely to maintain the same score, given that most of their concerns (feature-matched baselines, ablations, directed variants, and hyperparameter behavior) were directly addressed in the rebuttal.

---

### Decision · Program_Chairs · 2026-01-26

Reject